



# Performance enhancements on wind turbines using flow controllers: A review

Kiarash Kord.[a], Amir Noori. [b], Nahid Hasanpour.[c*], Ali Heydari.[d], Homayoun Askarpour.[e], Mehdi Kashfi.[f], Shayan Pakravan. [g]

[a] Dept. of Mechanical Engineering, K. N. Toosi University of Technology, Tehran, Iran

[b] Dept. of Mechanical Engineering, Shahid Rajaee Teacher Training University, Tehran, Iran

[c] Dept. of Energy and Physics, Amirkabir University of Technology, Tehran, Iran

[d] Faculty of Mechanical Engineering, Tarbiat Modares University, Tehran, Iran

[e] Dept. of Mechanical Engineering, Iran University of Science and Technology, Tehran, Iran

[f] Dept. of Mechanical Engineering, University of Tabriz, Tabriz, Iran

[g] Dept. of Mechanical Engineering, Sharif University of Technology, Tehran, Iran.

*Correspondence to:* Nahid Hasanpour (nazaninhsnpr@gmail.com)

**Abstract**

An effective strategy for enhancing the performance of wind turbines has been identified in the utilization of flow controllers. In this review paper, recent advances in flow control techniques and their impact on the aerodynamic performance of wind turbines at different Tip Speed Ratios (TSRs) are summarized. Emphasis is placed on the examination of the effects of various geometrical parameters of flow controllers on the performance of wind turbines through numerical studies. The aerodynamic principles underlying each flow controller and their functioning on wind

turbines are elucidated for each flow controller. The geometrical parameters of each flow controller, as explored in prior studies, are the focal point. The TSR range in which each technique is most effective, along with the advantages and limitations of each, is identified. Furthermore, the current state of knowledge concerning the utilization of active flow controllers and their impact on wind turbine performance is discussed. In conclusion, this comprehensive review paper presents an overview of state-of-the-art research on flow controllers for wind turbines, with a specific focus on

the geometrical parameters and TSRs studied, and identifies potential areas for future research and development.

**Keywords**: wind turbine, flow controller, geometrical parameter, performance enhancement

## 1. Introduction

Wind energy is a promising and growing renewable energy source that can help reduce reliance on fossil fuels.

However, wind turbines can be expensive to install and maintain, and optimizing their efficiency is an ongoing challenge. A study by the National Renewable Energy Laboratory (NREL) found that increasing the efficiency of wind turbines by just 5% can reduce the cost of wind energy by up to 9% (Mone et al., 2017). Additionally, by generating more energy from each turbine, fewer turbines are needed to meet the same energy demand. This can help reduce the environmental





impact of wind energy, as shown by the EWEA (European Wind Energy Association) report, which indicates that
increasing the efficiency of wind turbines by just 1% can reduce the number of turbines needed by up to 2.5% (EWEA, 2015).

So, improving wind turbine efficiency can help reduce wind energy's cost and environmental impact and increase its overall adoption and use. Flow controllers are devices used on wind turbines to modify air flow around the blades to improve efficiency and performance. These devices can be either passive or active, manipulating the airflow to reduce
drag and turbulence and increase lift.

By studying flow controllers, we can better understand how they affect wind turbine performance and identify the most effective designs and configurations for different wind conditions and turbine configurations. Therefore, studying flow controllers of wind turbines is essential for the continued development and improvement of wind energy technology.

Passive flow controllers work without using any external power source. They are passive because they do not require moving parts or energy input to operate. The purpose of these flow controllers is to reduce the aerodynamic loads on the wind turbine blades, which can help increase the turbine's lifespan and reduce maintenance costs. They can also improve the turbine's overall efficiency by reducing turbulence and improving the blades' angle of attack. They are becoming increasingly popular in the wind energy industry because they can be easily retrofitted to existing wind turbines and
provide significant performance improvements with relatively little cost or maintenance requirements.

On the other hand, active flow controllers work using an external power source, typically electricity. Unlike passive flow controllers, which do not require any energy input, active flow controllers can actively manipulate the airflow around the blades using sensors, actuators, and control systems. Their mechanism is mainly optimizing the angle of attack and reducing turbulence.

Passive flow controllers have lower costs, a more straightforward design, and lower maintenance requirements. Moreover, they have limitations, such as a limited range of effectiveness and the potential for increased noise levels. However, active flow controllers have greater control over airflow and can adapt to changing wind conditions. They also require more complex control systems and maintenance. Further research is needed to optimize for different wind conditions and turbine configurations. The location and configuration of different flow controllers on a blade are
illustrated in Fig. 1.

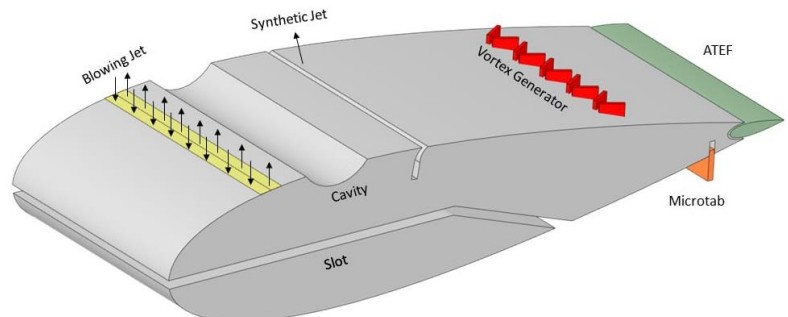





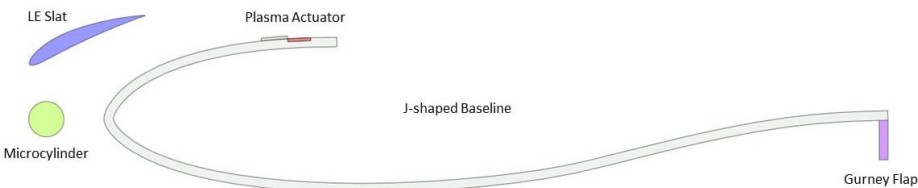

**Fig. 1. The location and configuration of different flow controllers on a blade**

Several studies have shown that the performance of wind turbines can be significantly improved by optimizing the configuration and geometry of flow controllers. Studying the configuration and geometry of flow controllers is also crucial for designing more efficient and cost-effective wind turbines. By understanding how different flow controllers work and how their configurations and geometries affect wind turbine performance, researchers and engineers can develop more effective and efficient flow controllers that can be integrated into wind turbines. In addition, studying the configuration and geometry of flow controllers can help improve our understanding of the complex aerodynamic flow patterns around wind turbines. This can lead to the development of more accurate and reliable computational models for predicting wind turbine performance and optimizing wind turbine designs.

This present article aims to compile an extensive overview of the flow controllers used to increase wind turbines' power generation, including their effects on performance improvement, their performances with different geometrical parameters at each TSR regime, and other salient benefits, advantages, and disadvantages.

## 2. Passive Flow Controllers

### 2.1. Flap

Flaps, located on the trailing edge of a blade, can be either movable or fixed surfaces that play a crucial role in enhancing the aerodynamic performance by altering the airfoil camber. These flaps, categorized as either active or passive flow controllers, contribute to increased lift by modifying the pressure distribution along the chordwise direction (Abbott and Von Doenhoff, 2012). The two common types of passive flaps are the fixed trailing edge flap and the gurney flap, which will be explained in the following sections.

The fixed trailing edge flap is a control surface positioned at the back of a wind turbine blade, allowing operators to adjust the blade's aerodynamic characteristics. Deploying this flap offers several advantages. Firstly, it provides better control over lift and drag forces acting on the blade (Castaignet et al., 2014). By adjusting the angle of the flap, the lift force can be increased or decreased, optimizing the turbine's performance under different wind conditions. Additionally, the fixed trailing edge flap helps mitigate turbulence effects and reduces blade loads. When the wind is turbulent, the flap can be adjusted to lessen load fluctuations on the blade, enhancing the turbine's structural integrity and lifespan (Mansi and Aydin, 2022a).

Furthermore, deploying the fixed trailing edge flap expands the wind turbine's operational range by adapting to varying wind speeds, leading to increased energy production and efficiency.

Gurney flaps, depicted in Figure 1, are flat plates mounted perpendicularly to the airfoil's profile on either side of the trailing edge. Their primary purpose is to enhance performance and alleviate loads. They significantly improve the lift-



drag ratio and have a slightly positive effect on dynamic stall. Gurney flaps work by generating counterrotating wake vortices that energize the boundary layer and prevent flow separation. This delay in separation shifts the stagnation point downstream of the trailing edge, resulting in improved lift performance of the airfoil (Liebeck, 1978; Liu and Montefort, 2007).


Incorporating a Gurney flap can notably enhance the power coefficient at low tip speed ratios, where power production is typically low. For example, in a straight-bladed vertical axis wind turbine (SB-VAWT), a 16.47% increase in maximum power coefficient is achieved at a TSR of 2 when compared to the clean blade's maximum power coefficient of 0.333 at a TSR of 2.5. Power coefficient curves align closely when TSR is higher than 3 (Zhu et al., 2021).


Even slight changes in Gurney Flap geometry, such as height, width, and mounting location (inboard, outboard, or both), significantly affect the airfoil's performance. Several researchers have studied the geometric parameters of Gurney flaps for airfoils (Basualdo, 2005; Jacob, 1998).

For instance, Yan et al. (Yan et al., 2019a) conducted a numerical study to investigate the effects of a Gurney flap on the aerodynamic performance of the NACA0018 airfoil. They considered five different heights ranging from 1 to 5 percent of the chord for the inboard-GF in conjunction with a three-blade rotor of an H-type Darrieus wind turbine. Zhu et al. numerically investigated five different heights ranging from 0.75 to 1.75 percent of the chord for the inboard GF in NACA0021 associated with a VAWT(Zhu et al., 2021). Another study on this specific airfoil and turbine examined GF heights of 2%, 3%, and 4% of the chord. The findings revealed that thrust increases with an increase in Gurney flap height, especially at lower tip speed ratios. Higher GF also increases the airfoil lift coefficient (CL) with a slight increase in drag coefficient (CD), which has a minimal effect on CL/CD. However, there is an optimal height beyond which the performance decreases. For example, the best configuration was achieved with a 2% chord Gurney flap installed on the inner side of the NACA0021 airfoil (Bianchini et al., 2019).



Md Farhad Ismail introduced a combination of GF and dimple and considered GF heights ranging from 0.25% c to 3%c on a NACA-0015 airfoil used in VAWTs. The study aimed to find the optimized configuration using Response Surface Approximation (RSA) to maximize the average torque produced by the wind turbine blade. The figure below illustrates the GF heights that were numerically investigated (Ismail and Vijayaraghavan, 2015).


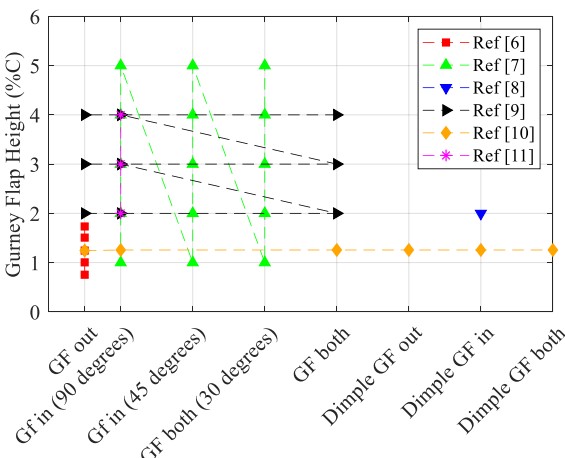

**Fig. 2.** Studied geometric parameters

GF could be mounted inboard, outboard, or on both sides of the airfoil at different angles (see **Fig. 3**). Numerical results show that the best performance witnesses at a 90-degree angle GF (Yan et al., 2019a).

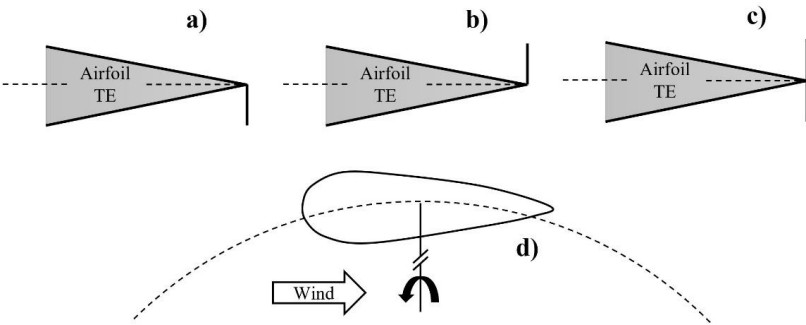

**Fig. 3.** Different locations for mounting Gurney Flap

This particular Gurney flap (GF) configuration effectively prevented the initiation of vortex shedding during high-

TSR rotational regimes, and this effect was even more pronounced when the flap was positioned on the outer side or on both sides of the airfoil. On the other hand, when the GF was mounted on the inner side of the blade compared to the baseline configuration, it resulted in a flatter power curve trend with a lower peak TSR.

Bianchini conducted research and found that by applying the GF on the inner side of the NACA0021 airfoil for Darrieus vertical axis wind turbines (VAWT), the power coefficient increased by 21.3% at TSR = 2.4 [11]. Similarly,

Zhu et al. demonstrated that three types of GF (outboard GF, two-side GF, and inboard GF) significantly improved the power coefficient of straight-bladed vertical axis wind turbines (SB-VAWT) at low TSR. The enhancement of the $C_P$ value with outboard GF, two-side GF, and inboard GF was 37.9%, 28.4%, and 23.7%, respectively, at TSR = 2.35 [7]. The mentioned information can be seen in Fig. 4.

Dighe et al. showed that installing a higher height flap significantly impacts the aerodynamic performance of the
diffuser-augmented wind turbine (DAWT) after evaluating two heights of 2% and 4% (Dighe et al., 2017).

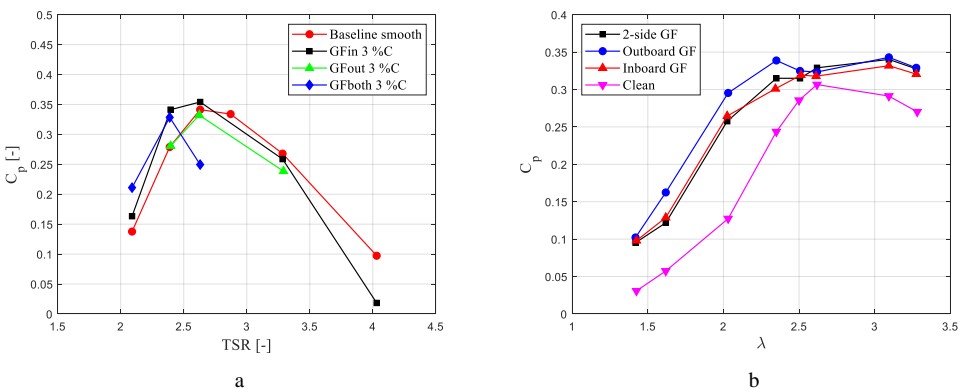

a                                             b

**Fig. 4.** Power coefficient of the entire three-blade case-study rotor. a) **NACA0021 - Darrieus Vertical-Axis Wind
Turbines** (Bianchini et al., 2019)**. b) NACA 0021 - SB-VAWT** (Zhu et al., 2021)

Haitian Zhu et al., by considering three different GF widths (0.04%c, 0.08%c, 0.12%c) for SB-VAWT with three
NACA0018 blades, demonstrated that the effect of GF width on aerodynamic performance is not apparent. However, a
wide GF is more suitable for SB-VAWT than a narrow one, especially at low TSRs(Zhu et al., 2021).

### 2.2. Microcylinder

Microcylinders are counted as an off-surface passive flow controller similar to the rods near the airfoil leading
edge(Jacob et al., 2005). The primary purpose of these elements is to delay separation, which is made possible by vortex
shedding interacting with the boundary layer and adding momentum to the near-wall region. The microcylinder diameter,
distance to the airfoil surface, and location (see Fig. 5) can sensitively affect the optimal configuration investigated by
Luo et al.(Luo et al., 2017) in 2017, considering NACA 0012 and flow field ranging from 16 to 23 degrees. However,
investigating 15 locations reveals that the optimal performance varies from case to case; the microcylinder could
effectively diminish the extent of the separation region resulting in a lower amplitude of aerodynamic forces on the
suction surface and thus delaying the heavy stall, as depicted in Fig. 6.





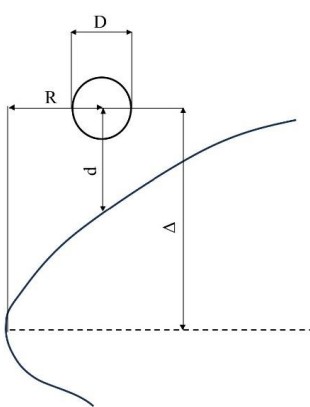

**Fig. 5. Studied geometrical parameters of microcylinders** (Wang et al., 2018)

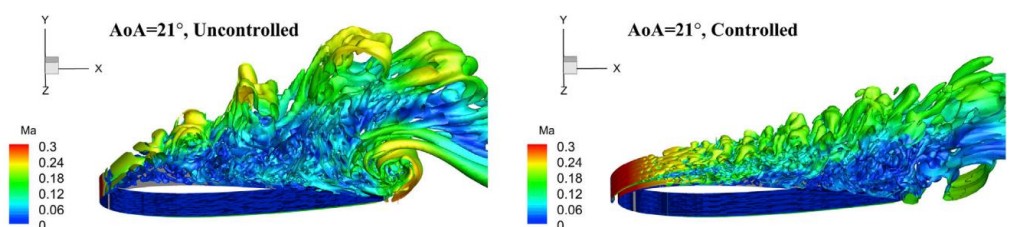

**Fig. 6. The iso-surface of controlled and uncontrolled airfoil colored by Mach number at AOA of 21 degrees** (Luo et al., 2017)

Wang et al. (Wang et al., 2018) investigated the effect of employing a microcylinder in front of a horizontal axis wind turbine blade. Their results also backed up the previous study and showed an improvement of up to 27.3% in wind turbine torque. At the light stall, a smaller diameter and more considerable vertical distance from the airfoil leading edge

(Δ) show a more significant improvement in torque. At the deep stall, larger diameter and more minor Δ indicate better results on torque improvement. In a recent study, the effect of a microcylinder on the performance of a VAWT based on the NACA0018 airfoil was explored (Bakhumbsh and Mohamed, 2022). The results revealed that smaller diameters improved the power coefficient more than larger diameters. Also, inserting the microcylinder in front of the airfoil introduced higher performance than when the microcylinder was inserted in the suction side of the airfoil. Moving along

the blade spanwise (R) has almost no effect on torque improvement. Moreover, Shi et al. (Shi et al., 2021b) proposed an innovative study and introduced an oscillating micro-cylinder upstream of the airfoil suction surface. The results show that the impact of oscillating microcylinders on aerodynamic performance highly depends on the initial position and oscillating mode, including amplitude and frequency. Generally, the oscillating microcylinder can help increase the lift-to-drag ratio by 88.21% when the optimal position and oscillating mode are given. To conclude, microcylinders are a

newly proposed flow-controlling method that must be studied in-depth because they show promising initial results and are a practical solution to delay the stall.





### 2.3. cavity

Cavities are geometrical modifications viewed as grooves carved on the surface of an airfoil(Zhu et al., 2018a). Olsman et al. (Olsman and Colonius, 2011) used 2D DNS and investigated flow behavior for different angles of attack

for a NACA0018 airfoil with a cavity placed on its leading edge at Rec=2E+4. Their study revealed that the cavity could generate vortices in the direction that energize the shear layer, dampens the separation bubbles, and hence delay the separation point downstream of the airfoil.

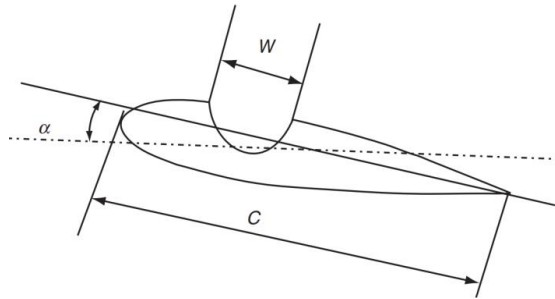

**Fig. 7. The shape of an airfoil with a cavity** (Vuddagiri and Samad, 2013)

The research ideas around cavities are mainly about the location of cavities in terms of the chord length and optimization of their geometrical parameters such as diameter, opening, depth, and sharp or chamfered edges. In the study performed by Yousefi Roshan et al. (Roshan et al., 2021), different configurations and shapes of arc cavities were extensively explored, and the effects of parameters such as cavity diameter, position, shape, and dual cavity mode were

investigated over torque and pressure coefficients of a Darrieus based on different tip speed ratios (TSR). Among the obtained results, it is worthwhile to mention that they concluded that the highest pressure and torque coefficient values were obtained when the cavity was placed on the suction side and near the trailing edge. They also mentioned that the dual-cavity mode increased the power coefficient by 17% at TSR=3.5 using the Upper and lower surfaces (trailing edge) mode.

Vuddagiri et al. (Vuddagiri and Samad, 2013) have investigated the effect of sharp or chamfered edges of a circular cavity placed on the leading or trailing edge of NACA0018 and NACA0024 airfoils, as depicted in Fig. 7. Their results showed that the stall angle increases from 14 to 20 for NACA0018 at Re=1E+6 when the cavity is placed near the trailing edge. However, regarding the use of cavities on the leading edge for a NACA0024 at Re=1.5E+5, they demonstrated that though the separation point was delayed from 60% of the chord length to 68%, the lift-to-drag ratio decreased,

compared with the clean airfoil. They also delineated that cavities with sharp edges show improved lift and drag coefficients compared to cavities with chamfered edges due to the vortices being well-confined inside cavities.

An optimization study on Risø_B1 airfoil was carried out using the genetic algorithm and considering 16 different shape parameters (Nili-Ahmadabadi et al., 2020). The optimization is based on an angle of attack of 14 degrees where the stall point occurs for the Risø_B1 airfoil, and it was aimed to delay the stall point.

Parametric simulations were performed to investigate the arc and rectangular cavity cross-profile shapes for the NACA4415 airfoil (Liu et al., 2020). The results showed that the most prominent factor for a cavity shape is its recess



depth ratio (h/δ, h: groove depth, δ: baseline boundary thickness), and its most effective value is between 1-1.5. they also investigated the location of the cavity in terms of chord length and concluded that the optimum end point of a cavity is located near 16% of the chord length. Finally, the results showed that a rectangular groove with h/δ between 1.2-1.5

performs better than an arc cross profile in efficiently confining flow streams within the cavity.

  In the study by Yadav et al. (Yadav and Bodavula, 2022), the effect of a triangular cavity on the aerodynamic performance of a NACA0012 at Re=5E+4 was explored. They investigated triangular cavity profiles on the suction side at 10%, 25%, and 50% of the chord length, with depths equal to 2.5% and 5% of the chord length. The obtained results showed that the triangular cavity with a depth of 2.5% of the chord length located at 10% chord length enhances the

aerodynamic performance by 52% at an angle of attack of 8 degrees. Also, it improves the aerodynamic performance by 10 and 17 percent when placed at 25 and 50% of the chord length. The deeper cavity also improves the aerodynamic performance by up to 13%, 22%, and 14% at angles of attack between α = 6 and 10 degrees. Another interesting point in this study is using the transition turbulence model, γ–Reθt, to capture the turbulence effect better. Regarding the use of turbulence models other than RANS, Błoński et al. (Błoński et al., 2021) employed the high-order penalized vortex

method to solve the flow passing over a Risø airfoil with two types of optimized cavities at Re=2E+4 and angles of attack of 3,6 and 9 degrees.

### 2.4. Vortex Generator

  Vortex generators (VGs))Taylor, 1947( are small fins that are installed over airfoils or toward the root of the wind turbine blade (see Fig. 1). VGs are passive flow control devices that prevent and delay flow separation by generating

vortices along the flow stream. They enter high-momentum flow into the boundary layer that opposes the reversed pressure gradient)Astolfi et al., 2018b, a(.

  The VGs application range is so vast that not only can they be used for low-Re cases, such as turbine blades (Moon et al., 2021) but also, they can be exploited for high-Re cases, such as in the transonic airfoils (Forster and White, 2014a). Although VGs are similar to each other in terms of application, their geometry may be different. Different types of VGs

have been introduced under the names of vanes (co- or counter-rotating delta, rectangular, trapezoidal), wheelers (wishbone and doublet), ramps (backward/forward), and many others (Lin, 2002) (see Fig. 8).

  Many studies focused on geometric parameters (such as length, skew/pitch angles, their location regarding chord length, and …) of conventional VGs, while some tried to introduce new and creative VG shapes(Zhu et al., 2018b). Numerical studies were carried out to design and optimize the geometry of VGs to obtain the best aerodynamic

performance for different airfoils at different Re numbers (Forster and White, 2014a), (Dandois et al., 2010; Yan et al., 2019b; Zhao et al., 2015). Most of these studies focused on a single airfoil (a limited span of a wind turbine blade or a wing) to acquire the maximum information regarding the VGs with the least possible computational cost (Wang et al., 2017; Zhao et al., 2015).

  In addition to static VGs, whose height is fixed, two types of VGs with the names of Smart and High-frequency Micro

Vortex Generators (SVGs, and HiMVGs, respectively) are also studied, with varying heights. These types of VGs have the advantage of eliminating/reducing the static VGs' parasitic drag by changing height. While SVGs can protrude from their casing on the airfoil to delay the separation point (Barrett and Farokhi, 1996), the HiMVGs oscillate with significant frequencies to produce intermittent vortices (Osborn et al., 2004).





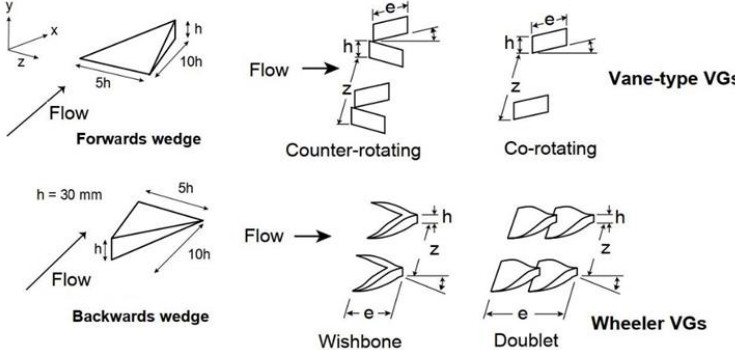

**Fig. 8. Conventional VGs' shapes** (Lin, 2002)

The most prominent factor for designing VGs is their location concerning chord length (x/c) (Dandois et al., 2010; Wang et al., 2017; Zhao et al., 2015). This parameter has a determining role in the optimum performance of VGs and was investigated in every numerical study ever carried out. It is shown that for AOAs in the range of ~ [10º - 20º], the airfoil's aerodynamic performance can be highly improved by placing VGs at 15-20% of the chord length (Wang et al.,

2017; Zhao et al., 2015). It was also shown that placing the VGs both so far downstream or upstream will result in reduced aerodynamic performance. If the VGs are placed downstream, the boundary layer thickness is relatively high, and the produced vortices are not strong enough to make the flow stay attached to the airfoil's surface. Also, the generated vortices will dissipate until they reach the separation point if the VGs are placed so far upstream of the airfoil (Wang et al., 2017).

The following important parameter is VGs' height which is designed based on the boundary layer thickness. Their height can be larger or smaller than the boundary layer thickness (Yan et al., 2019b). However, most of the VGs are designed so that their length would respond to boundary layer thickness at their respective location (Dandois et al., 2010; Wang et al., 2017). Still, it is shown that increasing the VGs height can cause more lift force to be produced (Gao et al., 2015). The reason can be attributed to the VGs' trailing edge height that causes high-momentum flow from upper levels

to be entered into the boundary layer, making the flow stay attached to the surface, while the airflow entering the o boundary layer using shorter VGs have less momentum. However, increasing the VGs' height will come with a penalty which is the increase in the generated drag force. Therefore, the aerodynamic performance curve should be noticed to justify the increase in VGs' height. Nevertheless, a type of VG has been developed with heights smaller than the boundary layer height at its respective location (Martinez Suarez et al., 2018). These VGs are generally used for low Re

numbers and are known as micro vortex generators (MVGs) (Akhter and Omar, 2021; Lin, 2002). They are primarily exploited due to their lower parasitic drag. Although these VGs impose lesser drag force on the system and may be considered a better alternative, the vortices they produce are weaker. Therefore, their location regarding chord length must be precisely chosen to guarantee the best performance (Akhter and Omar, 2021). Similar statements can be proposed for the VGs' length. Most reviewed numerical studies have considered this parameter fixed, and its effect has

yet to be studied extensively. Nonetheless, in the study by Gao et al. (Gao et al., 2015), two different VGs with lengths of 17 and 20.4 mm were studied. Their results showed that increasing the VGs' length from 17 mm to 20.4 mm does not considerably affect the generated lift force. They only caused the parasitic drag force to be increased drastically, resulting





in reduced aerodynamic performance (Table 1).

**Table 1. main parameters of VGs investigated in Gao's research** (Gao et al., 2015)

| Case no. | H (mm) | L (mm) | a (mm) | b (mm) | β (º) |
|---|---|---|---|---|---|
| VGs1 | 5 | 17 | 10 | 25 | 16.4 |
| VGs2 | 6 | 17 | 10 | 25 | 16.4 |
| VGs3 | 6 | 17 | 12 | 30 | 16.4 |
| VGs4 | 6 | 20.4 | 12 | 30 | 16.4 |

Regarding the other effective parameter for VGs, spacing, the results of the reviewed studies show that an optimum value exists for the spacing between VGs. Based on the obtained results by Gao et al. (Gao et al., 2015) increasing the space between VGs can improve aerodynamic performance. Dandois et al.(Dandois et al., 2010) also studied the effect of spacing between co-rotating VGs. Their results showed that if the spacing between VGs is smaller than a specific value ($\lambda/h < 6$, $\lambda$ being the space between VGs), the generated vortices would have a self-destructing state which causes

the aerodynamic performance curve to drop. They also investigated the effect of skew angle ($\beta$). Their results delineated that the maximum lift force is generated for the cases where $\beta$ is in the range of [10º - 20º] about flow stream direction.

### 2.4.1. Novel VGs shapes

Although the parameters mentioned earlier were studied for conventional VG types, there are exciting studies

proposing novel models for VGs that can be used to design the most optimum VGs in terms of aerodynamic performance. Different and novel types of VGs were introduced with different purposes, including VGs which combined the ramp and wheeler VGs (Forster and White, 2014a) with the target of improving the performance of each VG, a novel type of VG with the name of Rod Vortex Generator (Martinez Suarez et al., 2018; Suarez et al., 2016)with simpler geometry and with the target of decreasing the parasitic drag force applied over the airfoil, or even VGs inspired by nature and based

on the shapes of manta ray fins which improved the optimization of blade aerodynamic performance up to 10% higher than conventional delta vanes under different roughness conditions (Zhang et al., 2020a) (Fig. 8). Regarding the design of novel VGs, some studies suggested using double-row row conventional VGs to improve aerodynamic performance. Wang et al. (Wang et al., 2017)and Zhu et al. (Zhu et al., 2018b) used two rows of VGs and compared the caused effect with single VGs. Wang et al. showed that placing double VGs at 10% and 40% of the chord length could increase the

generated thrust and torque by 25% and 45% compared to the single-row VGs case(Wang et al., 2017). However, Zhao et al. showed that for unsteady aerodynamic responses, the double-row VGs effect is, in fact, the combination of single-row VGs in different locations (Zhu et al., 2018b).

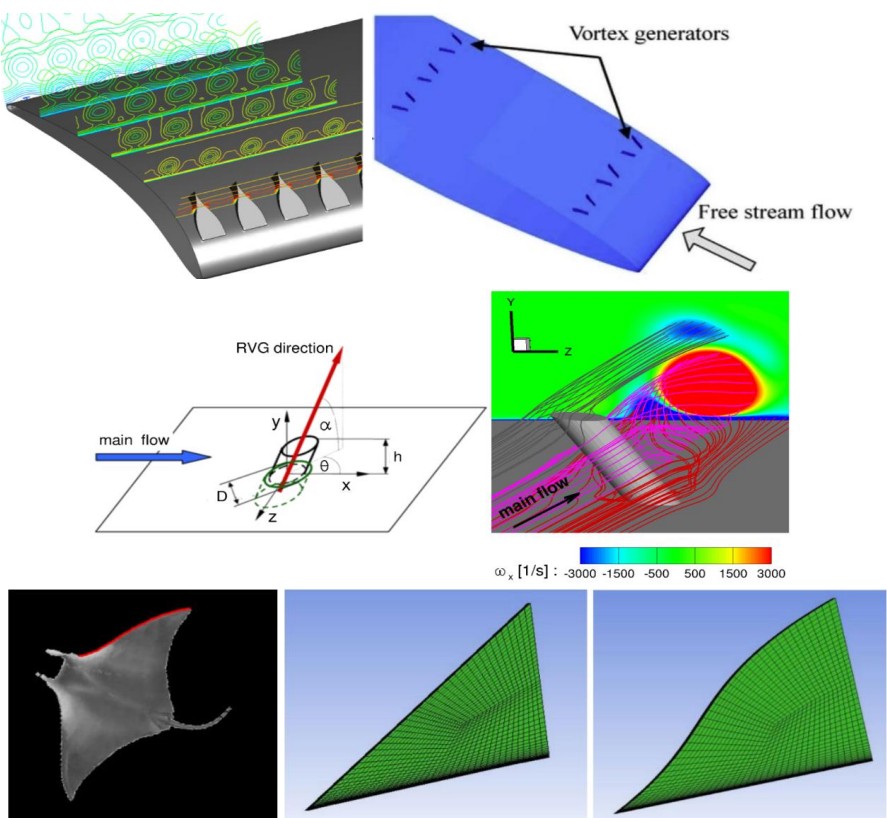

**Fig. 9. Novel VG shapes. a) combination of ramp and doublet wheeler VGs for exploiting merits of each** (Forster and
White, 2014b)**, b) double VGs to improve the aerodynamic performance compared to single-row VGs** (Wang et al., 2016)**, c)
Rod VGs used to reduce the parasitic drag due to simpler geometry** (J Martinez Suarez, P Flasz ńy ski, 2016)**,** (Suarez and
Doerffer, 2018)**, d) bionic VGs based on the shape of manta ray fin to improve the overall aerodynamic performance** (Zhang
et al., 2020b)**.**

**2.5. J-type blade**

The starting torque of straight blade vertical axis wind turbines (SB-VAWT) is one of the biggest concerns because
they are counted as lift-based wind turbines in contrast with Savonius wind turbines (type of VAWT). Thus, Zamani et
al. (Zamani et al., 2016a) came up with an idea and proposed a new cross-section called J-type, a combination of
Savonius and Darrius (see Fig. 10). It is designed to eliminate a fraction of the pressure side of the baseline profile Du
06-W-200. Using a j-type profile, the wind turbine would be driven simultaneously by drag and lift force, improving
starting torque. Fig. 11 illustrates that the power coefficient increases up to nominal TSR and then begins to decrease
but still has a higher power coefficient than conventional airfoil below TSR 2.5. The reason behind this is the increase
in the duration of dynamic stall peak over the upstream region and the overall increase in thrust loading on the
downstream side.



Furthermore, another study is conducted to simulate a J-shaped Darrieus vertical axis wind turbine with NACA 0015
baseline in three-dimensional that can help to comprehend the flow behavior (Zamani et al., 2016b). The rotor is divided
into four regions, including upwind, leeward, downwind, and windward, as displayed in Fig. 12 In the windward region,
when the angle of attack is equal or close to zero, there is not much lift force due to a symmetrical inherited baseline and
a small drag force is exerted on the airfoil. By increasing of azimuth angle, the lift force increases, and in the leeward
and downward regions, the drag force is captured by the cup-shaped section, assisting the rotation of the rotor.
Additionally, as the vortices form at the suction side inside the cup-shaped section, the crossing flow passes more
smoothly over the suction side resulting in less vortex shedding. Thereby, less vibration and noise are produced in the
wake region at the rear of the turbine.

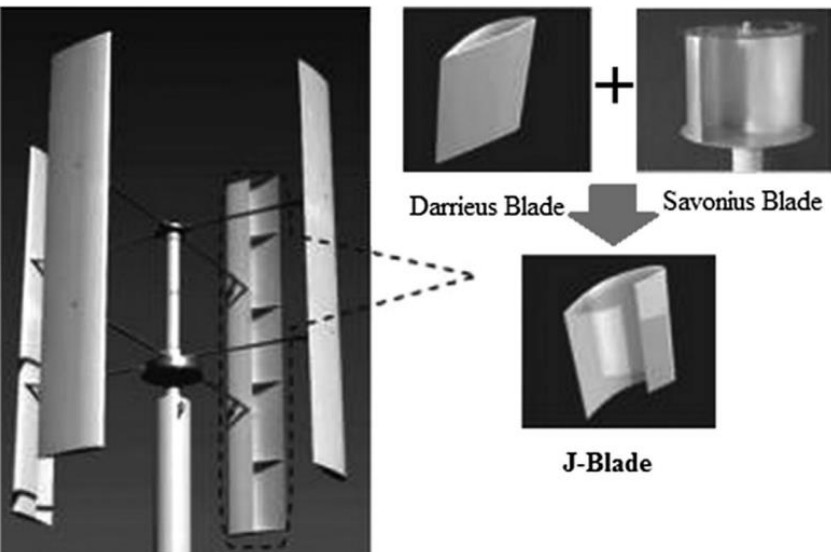

**Fig. 10.** *J-shaped cross-section* (Zamani et al., 2016b)




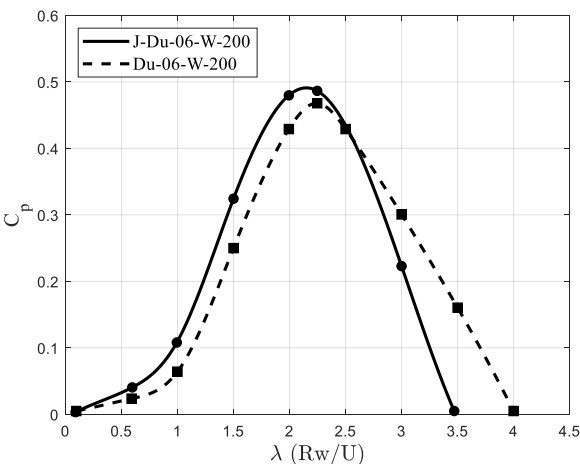

**Fig. 11.** *Power coefficient versus TSR* (Zamani et al., 2016a)

Interestingly, the idea of a J-shape Darrieus vertical axis wind turbine was challenged by Mohamed (Mohamed, 2019) in 2019 when he published a critical study and evaluated it from the performance and aero-acoustics viewpoint. A
numerical study tested three standard airfoils and twelve J-shape designs with different cut ratios and questioned their effectiveness based on its results. In 2021, Pan et al. [50] recently optimized a J-shaped blade on an offshore vertical axis wind turbine with geometrical correction (see Fig. 13), significantly improving upwind torque and wind energy capture rate. The convincing results necessitate the researchers to move forward and concentrate on new investigations. In 2022, Celik et al. Performed an extensive study to analyze J-shaped airfoils considering various design parameters.
The results have once again proved why J-shaped profiles should be counted as a possible solution to enhance torque generation and self-start ability(Celik et al., 2022).

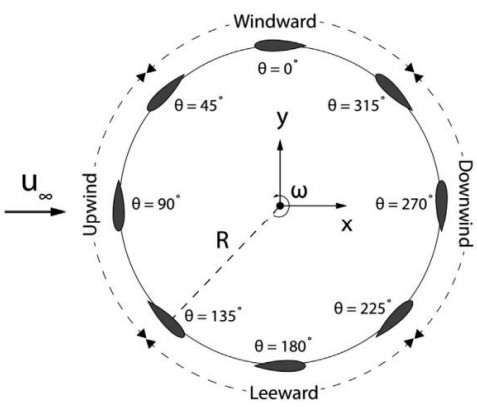



**Fig. 12. Schematic of four regions of the rotor** (Zamani et al., 2016b)

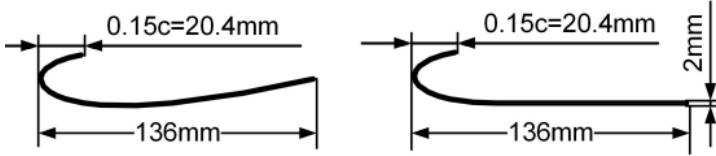

**Fig. 13. *J-shaped profile and optimized blade*** (Pan et al., 2021)

### 2.6. Leading Edge Slat (LES)

Numerous researchers examine the literature regarding implementing a new device to delay leading-edge separation as passive flow control. It is believed that leading-edge slats can aid in increasing flow kinetic energy by injecting high-momentum flow through the gap between the main airfoil and slat (see Fig. 14) to the boundary layer, thereby moving

ahead the separation point, similar to what vortex generators do (Prandtl, 1928).

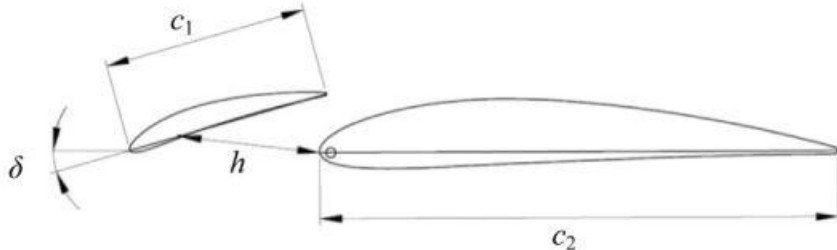

**Fig. 14. The geometry of an airfoil with a leading-edge slat** (Yavuz et al., 2015)

In recent years, several studies have explored the effects of position, angle of attack, and slat angle on airfoil and wind turbine performance. Chen et al. showed that the leading-edge slat has a more significant effect for a larger angle of

attack. As mentioned, the trailing vortex plays a vital role in the flow control process because the trailing vortexes of the leading-edge slat can affect the boundary layer kinetic energy. As a result, the flow on the suction side could resist the reverse pressure gradient and delay the separation phenomenon. Although the vortices generated by leading-edge slats at different installation angles differed, the leading-edge function decreased as its installation angle decreased. The aerodynamic performance should be analyzed in-depth considering different parameters, including lift and pressure drag

coefficient, x-velocity distribution in the boundary layer, and the torque and thrust coefficient done by Wang et al. (Wang et al., 2019). Their results prove that the leading edge could shift the separation point from $x/c=0.47$ to 0.67 and increase the lift coefficient by 52.99% in a case. Moreover, more studies focus on using slats on the leading edge of an airfoil of a horizontal axis wind turbine and vertical axis wind turbine to investigate the performance and seek optimal geometrical parameters (Ullah et al., 2020; Zaki et al., 2022). The results indicate that not only it can suppress the separation and

improve aerodynamic performance, but it may also lower the cost of the blade by reducing the used material in a clever design.

Furthermore, the slat position allows us to benefit from combining other passive flow control devices to enhance flow conditions on the surface and pressure simultaneously, as Li et al. (Li et al., 2022a) creatively proposed to combine the leading-edge slat and micro tab. Thus, the leading-edge slat can increase the flow energy on the suction side of the airfoil,



and the micro tab can hinder the flow of the pressure side. In another novel study, Kumar et al. (Kumar et al., 2021) employed multiple slats and used a multi-objective genetic algorithm (Golberg, 1989) to optimize the position of the secondary slat. The additional slat improved the overall aerodynamic performance and delayed the separation to the higher angle of attack by reducing the boundary layer thickness.

**2.7. slot**

The slot is a passive fluid control method widely used to improve the aerodynamic performance of traditional vertical and horizontal axis wind turbines (Zhang et al., 2022).

The blade's slotted design (see Fig. 15 (a)) creates an internal flow that increases air velocity as it exits the slot and impacts the incoming air at the bottom portion of the blade. High-velocity flow under the blade disrupts streamlined flow along the airfoil and separates flow from the surface, reducing fluid velocity. More lift is produced because of the

increased pressure under the blade due to the velocity reduction (Ibrahim et al., 2015). Compared with the conventional airfoil, it is observed that the airfoil with a slot has a higher lift and a higher lift-to-drag ratio at a large attack angle (Ni et al., 2019).

The leading-edge slot is a long, narrow aperture or slit on the airfoil's leading edge to draw some incoming airflow into the airfoil and push it out of the pressure side (Zhang et al., 2022). As depicted in Fig. 15 (b), geometrical parameters

of slot that have been investigated in recent literature are slot relative width (w), the angle between the slot's first leg and the horizontal line ($\beta_1$), slot vertical position (h), length of the first leg of the slot ($L_1$) and angle between the first and second legs of the slot ($\beta_2$) (Beyhaghi and Amano, 2018).

An investigation of slot relative widths indicated that for angles of attack greater than 6 degrees, the lift coefficient increased with a slight increase in drag coefficient while other parameters were constant. The effects of slot inlet angle

improved lift and drag coefficient in all AOAs using optimal slot relative width. The slot's vertical position was examined for three different modes. Based on the results, it was found that lowering the slot from its original position improves the performance of the airfoil (Beyhaghi and Amano, 2018). Also, a smaller slot with a longer length can offer better performance, due to Alexandrina Untaroiu et al. (Untaroiu et al., 2011) but there is a threshold for a slot thicker than 0.01 ft that does not remove enough mass.

Sercan ACARER has employed a multi-objective genetic algorithm to consider different ranges of geometrical parameters for slots to maximize the peak of $C_L/C_D$ 's airfoil.(Acarer, 2020) It is evident that all the parameters cause an increase in $C_L/C_D$, and $\beta_1$ has the most negligible influence, while $\beta_2$ and w have the most significant influence on $C_L$. He also carried out a 2D CFD simulation to investigate the effects of the enhanced slot on both HAWT and VAWT. Results indicate that the peak of Cp increased by 3.2% for HAWT and remained unchanged for VAWT; However, high

tip-speed-ratio ($\lambda$>3) Cp values increased between 3.5 to 9.6% throughout the VAWT operational range.

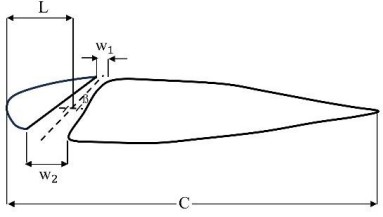 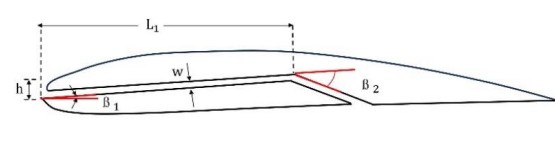



**a) Leading edge slot** (Beyhaghi and Amano, 2018)          **b) Internal slot** (Mohamed et al., 2020)

**Fig. 15.** *different slots*

Yonghui Xie et al. (Xie et al., 2013) showed that using an internal slot in an S809 airfoil would improve the lift-to-drag ratio at high AOA (15 and 20 degrees). Riyadh Belamadi et al. (Belamadi et al., 2016) confirmed previous observations and extended the numerical simulation in all ranges of AOA between 0 and 20. It should be pointed out that less efficiency of the slot at low AOA is due to increased drag forces by slot presence. Adding an active slot gate that enables closure at lower AOA can alleviate this issue.

They also studied the influence of slot position, width (w2/w1), and slope. The best location of the slot is upstream of the separation point that is changing at different AOA and for different airfoils. The slot's inlet is on the pressure side, and the outlet is on the suction side, so a larger inlet and smaller outlet cause a high-velocity outlet, improving the slot efficiency. Omar Sherif Mohamed et al. (Mohamed et al., 2020). Then the best configuration of slotted airfoil was used in Darrius-type wind turbine simulation. Results showed that the power coefficient has improved at low TSRs (<2.5), approximately three times that of the baseline turbine.

In another investigation by B. Navin Kumar et al. (Rajendran et al., 2020), a new method for braking systems of wind turbines by applying the chord-wise slot is introduced. It controls wind turbines over speeding at high wind speeds by changing pressure distribution over the blade lift decreases and drags increases.

## 3. Active Flow Controller

### 3.1. micro tab and microjet

Microtabs introduced by Yen and Van Dam (Nakafuji et al., 2001; Yen et al., 2000) as a practical active load control system are small translational tabs generally positioned on the airfoil's surface near the blade's trailing edge on either side. A micro tab whose height is on the order of the boundary-layer thickness alters the Kutta condition by modifying the camber, thereby shifting the stagnation point (Standish and Van Dam, 2005).

Chow and van Dam (Chow and Van Dam, 2006) have performed a numerical flow simulation around an airfoil with a deploying micro tab device. The instantaneous streamlines in the tab region are shown in Fig. 16. As soon as the micro tab deploys and the fluid flow collides, a low-pressure zone forms behind the tab and causes a clockwise vortex (Fig. 16 (b)). This creates a force close to the trailing edge downward, generating a nose-up moment. As a result, the lift reduces until the vortex extends enough to reach the airfoil trailing edge (Fig. 16 (d)). A part of the upper surface flow joins the lower surface vortex unexpectedly. The central vortex tries to push the joined flow upstream along the lower surface to reach down the tab's suction side to the tab's lower tip (Fig. 16 (e-f)). Finally, the two flows leave the airfoil's surface at this new stagnation point. Despite its simplicity and high efficiency, the delayed response has become a weakness for micro tab compared to the other active flow control systems.

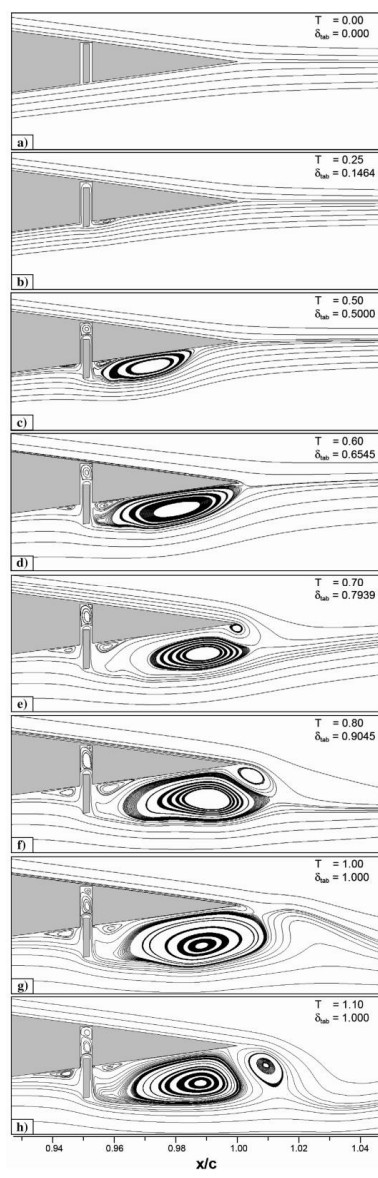

**Fig. 16. Streamlines during tab deployment (a-h) represent time stations** (Chow and Van Dam, 2006)

A first attempt to improve the efficiency was made by van Dam et al. through the conduction of sensitivity analysis to determine the appropriate micro tab configuration (van Dam et al., 2003). They selected a high-lift, low-drag GU25-5(11)-8 airfoil developed at the University of Glasgow (Galbraith, 1985; Kelling, 1968) and a wind turbine blade airfoil S809, developed at the National Renewable Energy Laboratory (NREL) (Somers, 1997; Wolfe et al., 1997). The main parameters under scrutiny were the tab height and location on the upper side (suction) and lower surface (pressure).



Later, J.P. Baker et al. (Baker et al., 2007) did a comprehensive study with the same purpose, using the S809 as a baseline profile. Both studies agreed that placing the tab on the lower (pressure) surface produces a lift increment while placing
the tab on the upper (suction) surface, causing similar pressure changes on the suction side, and lift is mitigated.

Moreover, the simulation of micro tabs with different heights and locations (Baker et al., 2007) demonstrated that tab implementation would be more effective with a more considerable tab height and closer to the trailing edge. For a fixed tab height, by implementing it far enough ahead of the trailing edge, the flow separation is not sustained with the flow and gets the chance to reattach, causing lift loss. Above all, the optimal location and height of the tab for lift enhancement
depend significantly on the geometric and aerodynamic characteristics of the baseline airfoil.

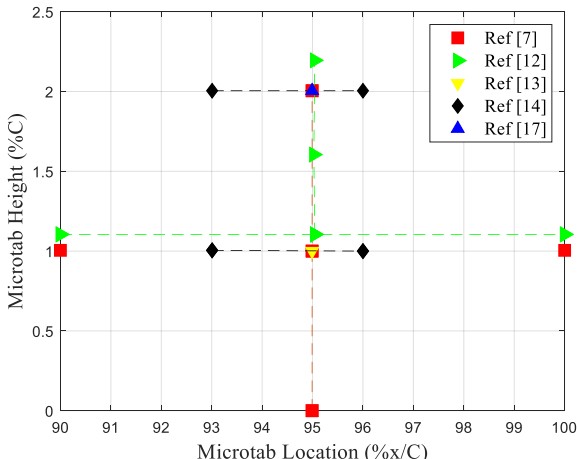

**Fig. 17. studied geometrical parameters of micro tab**

Another study presented the relationship between tab solidity and lift-to-drag ratios. The tab solidity ratio and increment in the lift are found to be highly linear. In other words, gap spacing causes an effect on pressure distribution
by weakening the suction peak and unloading the pressure side of the airfoil (Mayda et al., 2005). In conclusion, the gaps are estimated to decrease efficiency by 6-7% compared to a gapless tab spanning the entire model.

Fernandez-Gamiz et al. (Fernandez-Gamiz et al., 2017) performed a parametric study based on CFD calculations of a micro tab mounted on the pressure side of the DU91W (2)250 airfoil typically used on wind turbines. The results revealed that the average turbine power increased with implementing micro tabs on 5MW wind turbines.
Similarly, the effect of deploying micro tabs on performance improvement of a horizontal axis wind turbine blade was investigated in three-dimensional (Chen and Qin, 2017; Ebrahimi and Movahhedi, 2018). Their reported results emphasized the feasibility and efficiency of micro tabs on wind turbine performance. Moreover, the micro tabs have a simple structure, and can be used along with the other flow controllers to complement each other. For instance, Recently, the trailing-edge flap with micro tab has been exploited to enhance the performance of wind turbine blades(Ye et al.,
2021). The results showed a dramatic increase in the maximum lift coefficient by 25% and a delay in the airflow stall. The innovative idea proposed by Li et al. (Li et al., 2022b) of combining the leading-edge slat with the micro tab could also be mentioned. According to the numerical results, the combination could inhibit flow separation while enhancing





the aerodynamic characteristics of the S809 airfoil. In a single scenario, the lift coefficient was 171% greater than that of a clean airfoil, demonstrating its utility.

In recent years, researchers proposed discharging flow from an airfoil section normal to the surface to optimize the Cl/Cd ratio. Microjets or pneumatic jets are small jets near the trailing edge that blow air perpendicular to the airfoil's surface and alter Kutta's condition. However, the application of tangential blowers has been studied for decades (Hinton, 1957; Pearcey, 1961; Wallis, 1952; Wilkerson et al., 1974), and microjet's application is entirely different. The tangential blowers are used to change the momentum balance of the boundary layer to keep the flow attached to the surface and

delay the stall. In contrast, the pneumatic jets are used to create a vortex that changes the Kutta condition and affects lift force; depending on which side of an airfoil is used, it would be increased or decreased. The microjet and micro tab's effect on flow behavior over an airfoil surface is very similar, as depicted in Fig. 18.

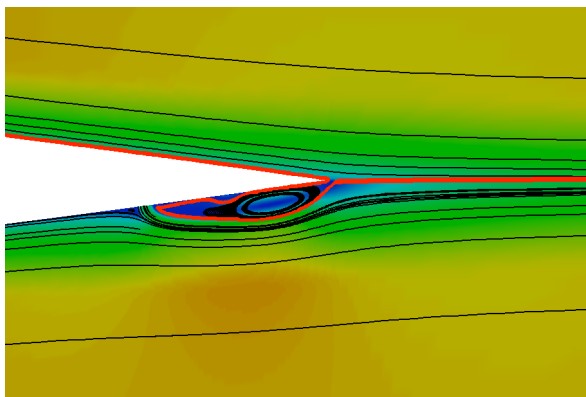

**Fig. 18. The instantaneous streak lines around the trailing edge region of an airfoil with a microjet on the pressure side**

(Brunner et al., 2012)

    Brunner et al. (Brunner et al., 2012) studies showed promising results and proved CFD's capabilities by comparing microjet wind tunnel tests with numerical results. Later, Blaylock et al. (Blaylock et al., 2014) performed a numerical study to make a meaningful comparison between pneumatic jets and micro tabs. Their study demonstrated that both affect flow around trailing edges similarly; therefore, the aerodynamic performances are very similar. Interestingly, the

microjets produce less drag force than micro tabs, and in some cases, drag increments were 30% lower for microjets while the lift and moment of both devices are very close. Although implementing a microjet instead of a micro tab seems more logical based on its aerodynamic performance, other factors like cost, ease of implementation, and working conditions should be considered before making a final decision.

**3.2. Blowing and Suction**

Regarding active flow control strategies for wind turbines, blowing/suction is the most popular option. Blowing and suction-flow control work by energizing the boundary layer to maintain an adverse pressure gradient and delay the stall angle. Injecting high-momentum flow into and inhaling low-momentum flow from the boundary layer works well to suppress or eliminate the flow separation, decrease drag, and increase airfoil lift.

    From a physical point of view, suction increases lift by producing a more extensive and lower pressure zone on the

top surface of the airfoil; as a result, the flow is more attached, and profile drag is decreased. On the other hand, leading-



edge blowing enhances lift by boosting circulation around the separation bubble and the airfoil, but at the expense of significantly higher leading-edge pressure; as a result, the flow becomes more separated, and profile drag rises, so it is typically ineffective at high wind speeds (Huang et al., 2004a; Tongchitpakdee et al., 2006). It should be pointed out that Huang et al. (Huang et al., 2004b) concluded that suction jet, compared to blowing, is dominant, even when the blowing

amplitude is six times the suction amplitude. The study was done on a NACA0012 at 18 degrees of angle of attack.

Numerous studies exist on blowing or suction active control in different conditions, such as single or multi-jet actuators, static or dynamic stalls, co-flow jet airfoils, Coanda effect design, etc. In this regard, Genç et al. (Genç et al., 2011), comparing single and simultaneous blowing and suction, observed that the best results were achieved by single-suction jets, multi-jets, and blowing jets, respectively. Also, For the no-flap condition, a single suction jet produced the

highest CL/CD value, and the separation point from the leading edge was delayed by about 47% (Evanovich, 2021).

Multiple actuators have been used to help increase the efficiency of wind turbines simultaneously. Although this will multiply the power required, increasing the output power by an appropriate amount can retrieve this cost. Kornilov et al. (Kornilov et al., 2019) studied singular slots on the opposite sides near the trailing edge at angles of attack from −6∘ to 6∘. Results show that blowing significantly affects the lift and drag coefficients of the airfoil, while the lift increase in

the suction case is more minor comparatively. On the other hand, increasing the amount of jet rate did not affect the aerodynamic performances significantly. However, when the blowing power rises, there is a risk that flow separation may develop close to the slot, causing a considerable rise in the drag force. Fatahian et al. (Fatahian et al., 2019) worked on a suction case on Naca 2415 airfoil with a high Reynolds number flow. On the airfoil's upper surface, nine slots with a width of 2.5 % airfoil chord length and an interval of 10 % airfoil chord length were placed independently. At lower

angles of attack, the lift coefficient was less sensitive to variations in the suction velocity ratio and the suction angle. However, at greater angles of attack, the influence of the suction application parameters was more apparent. Also, the optimum lift coefficient occurred when the slot location was between 0.3 to 0.6 of airfoil chord length. But the drag reduction was significantly affected when the slot location approached the airfoil's leading edge.

Sun et al. (Sun et al., 2020) realized that boundary layer suction could effectively enhance a vertical wind turbine's

performance and power coefficient. Applying a two-side boundary layer suction located at 30% chord length on both pressure and suction surface, at the optimum tip speed ratio (TSR) at TSR=2.33 for C=0.0075, the percentage increase in power coefficient on VAWT could reach up to approximately 34.2%. In another paper, Sun & Huang (Sun and Huang, 2021)found that when compared to single-suction, appropriate double chordwise suction slots can achieve higher power efficiency for both low and high TSRs. To optimize the simulation, they concluded that at low and high TSRs, suction

slots should be used at 10-30% and 30-50% airfoil chord lengths, respectively.

Sasson and Greenblatt (Sasson and Greenblatt, 2011) applied both single and double 45º injections at a 5% chord of a Vertical Axis Wind Turbine. While double-sided blowing may improve net annual energy production by over 150%, stalled blades saw a rise in net annual energy yield of over 60%. Based on these findings, there is a tremendous potential to increase the yearly energy output of the turbines by using active flow control blowing jets on VAWTs in an urban

environment.

Different parametric solutions of jet angle, jet length, jet-speed ratio, and location are carried out to review single jets of blowing or suction.

Suction and blowing configurations cross the boundary layer (CBL) and tangential to the boundary layer (TBL) are





mainly studied. It is concluded that cross-suction at the leading edge is better than other suction situations. Also,
tangential blowing at downstream places is better than other blowing places for boosting lift for the NACA0012
airfoil(Huang et al., 2004a).

Kamari et al. (Kamari et al., 2018) compared TBL and CBL blowing/suction effects on Selig–Donovan 7003 airfoil.
In a physical view, in all cases, in addition to inducing a delay at the beginning of separation, the TBL jet entirely
eliminated separation flow from the top side of the airfoil at both angles of attack (13 and 16 degrees), resulting in no
reversed flow downstream of the control jet location. From a parametric view, the lift increase in the TBL jet is more
effective than in other controlled cases. However, the best modification for the lift-to-drag coefficient is perpendicular
suction. Additionally, the optimum location for its application in all cases is close to the leading edge.

Serdar Genç & Kaynak (Genc and Kaynak, 2009) investigated NACA2415 at a transitional Reynolds number of 2e05.
At the same time, a single jet with a width of 2.5% chord is located on the suction surface at approximately 30% of the
chord. They concluded that regardless of the blowing locations, smaller blowing velocity ratios suppress the separation
bubble more effectively than larger blowing velocity ratios. Larger suction velocity ratios, on the other hand, are better
for suppression than smaller suction ratios.

Yousefi et al. (Yousefi et al., 2013) used a parametric solution to discover the optimum jet width and suction jet
entrance velocity, which is normal and uniform and positioned 10% of the chord length from the leading edge at
AoA=12-22 deg. This study found that when suction amplitude increased, the lift coefficient improved, and drag
coefficient decreased. Lift-to-drag ratio increased most at 0.5 m/s suction amplitude. Visbal (Visbal, 2014) calculated
blowing and suction on the leading edge of a NACA0012 airfoil for jet widths from 1.5% c to 4% c. As the blowing jet
width increased, the lift-to-drag ratio rose continuously in tangential blowing and declined quasi-linearly in cross-
blowing. Tangential blowing worked best with jet widths of 3.5–4% c, while perpendicular blowing performed better
with lower jet widths. The lift-to-drag ratio peaked at 2.5% c. Dighe and Cater explored distributed suction from x/c =
50% to 95% of the airfoil chord length at angles of attack from 2-16 degrees and five suction velocities (Dighe and Cater,
2015). The optimum result achieved at suction velocity=0.5 m/s for α = 6°. Flow control also lowers the airfoil's stall
angle by 4 degrees from 12 degrees and the lift-to-drag ratio grew as the suction jet length increased.

Perpendicular Suction is applied by Yousefi and Saleh (Yousefi and Saleh, 2015), at the leading edge of the wing's
upper surface, with two different types of slot distributions: i.e., center Suction and tip suction. With center suction,
more vortexes were eliminated than with tip suction, so the improvement in aerodynamic characteristics was more
evident. However, tip suction is preferable when the jet length is less than half the wingspan, whereas center suction is
superior when the jet length is higher than half the wingspan. Moussavi & Ghaznavi (Rezaeiha et al., 2019) studied the
effect of a boundary layer suction on a WENRI87 wind turbine in which the suction slot center with dimensions of 10m
and 0.05m is placed at 50% of chord length considering the limitations imposed by fixed blade geometry. They observed
a 5% increase compared to nominal power, while Flow reattachment was shown to be the primary cause of development.

In conclusion, when the suction jet width expanded, the lift-to-drag ratio also increased, peaking at 2.5% of the chord
length. Jet widths of 3.5% and 4% of the chord length for tangential blowing were the most effective, whereas lower jet
widths were more successful for perpendicular blowing. Also, the lift-to-drag ratio grew as the suction jet length
increased at both center and tip suction.

The effects of suction and blowing with and without flap are studied by Evanovich, J., & King, L (Evanovich, 2021).



They concluded that the flap positively affected the airfoil's lift, but high flap deflections negatively affected the drag coefficient, which decreased the lift-to-drag ratio.

Pulsed blowing, as opposed to a continuous fluid flow, transmits short pulses into the boundary layer and has been proven more effective. The S809 NREL wind turbine at Re= 2e06 and AoA=5 is investigated by Bobonea (Bobonea, 2012), while the pulsed blowing jet is located at 50% chord length with 45 degrees blowing. They overcome adverse pressure gradients and postpone separation by injecting stored high-velocity air into the boundary layer through slots. The effectiveness of pulsed blowing actuation is more robust in the 0.1-0.4% momentum blowing coefficient range. In comparison, the changes are more minor, from 0.4% to 0.5%. Another pulsatile actuator, TAU-0015 airfoil, is used by

Ekaterinaris (Ekaterinaris, 2004), and it is concluded that efficient flow control is possible for the weakly separated flow. Massively separated flow at high angles of the attack remained separated even with flow control, although the airfoil's performance was enhanced.

    The next subject focused on the simulations, in which the blades are pitching and, consequently, dynamic stall phenomena will be critical. Abdolrahim Rezaeiha et al.(Rezaeiha et al., 2020) , by investigating different tip speed ratios

and suction locations at various Reynolds numbers ranging from 0.51e05 < Rec < 2.78e05 on a pitching VAWT NACA0018 showed that suction applied along with the laminar separation bubble (optimally, at the LSB's downstream end) could prevent it from bursting, eliminate or delay its development, avoid the formation of the dynamic stall vortex and trailing-edge roll-up vortex, inhibit the production of the trailing-edge roll-up vortex, and delay trailing-edge separation. This all result in a significant increase in blade lift, and drag force reduction, beside a delay in stall angle.

Also, the optimum location of the suction slot is highly sensitive to the Reynolds number.

    2D and 3D (spanwise-non-uniform) simulations are carried out by Miguel R. Visbal (Visbal, 2014) while the airfoil is pitching at a constant rate. The blowing/suction jet is located on the airfoil pressure surface downstream of the leading edge. A significant delay could be seen in the early stages of a dynamic stall resulting in a higher maximum lift. In addition, both spanwise uniform and non-uniform actuation modes were efficient. In a comprehensive conclusion,

blowing/suction flow control for a light dynamic stall on pitch-up and down or sinusoidal movements could be quite effective.

    Tadjfar and Asgari (Tadjfar and Asgari, 2018) studied a blade oscillating between angles of attack of 5 and 25 degrees and observed the effect of jet location and jet velocity ratio with a tangential blowing jet. They located the jet slot upstream of the counter-clockwise vortex. They observed significant positive effects on lift and drag while placing it at

the separation point made inverse behavior of the drag hysteresis curve. Downstroke motion had much less drag in a controlled condition than upstroke motion, but the opposite was confirmed in the uncontrolled case.

    Asgari (Asgari and Tadjfar, 2018) simulated a similar case with pure continuous blowing and suction focused on the dynamic stall vortex as the main contributor to the drag increase resulting in an improvement in lift and drag and a reduction in the thickness of the separated region. Also, the dynamic stall vortex was suppressed in the blowing case and

eliminated in the pure suction case. Comparing both blowing and suction with each other, suction was superior to blowing in controlling the flow in the dynamic stall. Rezaeiha et al. (Rezaeiha et al., 2020) applied Steady and Unsteady Suction near the leading edge of VAWT NACA0018. They concluded that the dynamic stall on the turbine blade could be efficiently suppressed by a slot at the leading edge, along the chordwise length of the laminar separation bubble, and upstream of its bursting region. At various tip speed ratios of 2.5 and 3, a power gain of 247% and 83% was reported for



steady suction. Unsteady suction can also result in the same power gain in the dynamic stall, while the suction system's power consumption can be lowered by 65%.

Aeroacoustics noise reduction can also benefit from blowing-suction-flow control. Arnold et al. (Arnold et al., 2018a) simulated a generic NREL 5MW wind turbine in order to calculate the acoustic parameters of the solution affected by the boundary-layer suction system. As a result, there is a fantastic improvement in total rotor power throughout a broad

range of design states and a reduction in total trailing-edge noise. But as a clear crossover design point, the massively rising pump power demand outweighs the aerodynamic gains from suction and blowing. Confirming the above achievements, Arnold et al. (Arnold et al., 2018b) investigated a state-of-the-art N117 turbine to show the outcomes in an industrial setting resulting in up to 3.6 dB improvements above the baseline state, besides an enhancement in the aerodynamics of up to 4.75% of the primary rotor power. Furthermore, by a reduced level of 5 dB, the increase in

aerodynamic power exactly compensates for the pump power requirement; however, extremely high levels of overall noise reduction are possible only by reducing rotor power. To find out the best location of the blowing slot in noise reduction. Using both the forces of blowing and suction, Liu et al. [43] intended to enhance the aerodynamic performance of an H-type Darrieus wind turbine. By reducing pressure fluctuations, regulating the flow field, and controlling the vortex shedding, the active device is shown to diminish noise emission in an aeroacoustics noise estimate. Similarly, by

adjusting the sound pressure spectra between 100 and 1000 Hz, the suggested active control approach could reduce the noise output of wind turbines by as much as 6.56 dB(Liu et al., 2022).

Gerhard et al. (HAMID, 2011), using both LES turbulence simulation and measurements of the Somers S834 wind turbine, showed that at 90% of the chord length, the slot location was the most advantageous to significantly reduce the turbulence intensity and the induced surface pressure fluctuations in the trailing edge region.

In an innovative study, Niether et al. (Niether et al., 2015) designed a unique flow control that works depending on how the angle of attack affects the pressure differential. Next, one of the four outlet channels receives the actuation air. These direct the air to various chordwise exit points on the suction side of the DU97-W-300 airfoil. Significant order of reduction was seen in lift coefficient variation. Eventually, it was determined that, on average, less than 1% of the rated turbine power is required to power the flow control device.

**3.2.1. Coanda jet**

In the next step, the literature on the numerical Coanda effect on the performance of wind turbines is introduced. This actuation method tangentially injects high-momentum air into the boundary layer, particularly near the airfoil's curved or rounded trailing edge. The influence of the high-velocity jet improves the total airfoil circulation and sectional lift coefficient, which moves the stagnation point toward the pressure side. In the Djojodihardjo et al. (Gerhard et al., 2014)

study, modeling NACA2412 and S809 airfoil with a rounded trailing edge, the efficiency of Coanda-jet in generating enhanced lift, which could lead to higher torque in wind turbine applications, enhanced lift-to-drag ratio, and an increase of axial thrust for propeller applications, was apparent.

Kara et al. (Kara et al., 2013) saw beneficial increases in power generation when the wind speed was low or the blade pitch was raised to make the effective local angle of attack more minor. However, if the local angles of attack were high

enough to separate the leading edge significantly, the trailing-edge blowing proved useless.

Alexandru et al. (Dumitrache et al., 2014) modeled a GACC airfoil, one with a cylindrical trailing edge surface and a second configuration involving a blowing flap resulting that When compared to the baseline rotor, the use of trailing





edge blowing and the Coanda effect significantly increases the circulation around the airfoil section. However, when high wind speeds separate the flow, trailing edge blowing loses its effectiveness in boosting power production. Isac et al. (Isac et al., 2015), by simulating a Supercritical General Aviation Circulation Controlled Airfoil (GACC) with a cylindrical trailing edge, concluded that a significant rise follows the large lift increase in drag.

Petrucci et al. (Petracci et al., 2019) applied a Coanda jet with two different slot thicknesses and two jet momentum coefficients to the upper surface of an S809 airfoil from the leading edge to the trailing edge x/C=0.95. The results showed that high wind speeds make the jet useless, indicating that the Coanda jet is only functional when the flow is entirely attached or, at the very least when a slight separation around the stall point occurs. Additionally, the blowing jet, placed near the leading edge, reduces the stall angle and may be used for other systems, such as the de-icing system.

Mamou and Khalid (Mamou and Khalid, 2007) compared the cases of the Coanda jet and flap jet separately and simultaneously on WTEA airfoil-HAWT steady and time-accu. He found that the combined jet arrangement optimized aerodynamic performance for four different airfoil configurations (no-jets, Coanda jet only, jet flap only blown typically, and a combination of Coanda jet and jet flap).

### 3.2.2. Co-flow jet

The co-flow jet (CFJ) airfoil idea was recently created by Zha et al.(Zha and Paxton, 2004)  to enhance lift, decrease drag, and increase stall angle with little energy consumption. The suction surface of the co-flow jet airfoil has a suction slot close to the trailing edge and an injection slot close to the leading edge. An equal quantity of mass flow is sucked near the trailing edge, while a high-energy jet is tangentially injected near the leading edge. The main flow is able to overcome a significant unfavorable pressure gradient and remain attached at a high angle of attack owing to the significant turbulence diffusion and mixing caused by the turbulent shear layer that separates the main flow from the jet. In a numerical study on the CFJ0025-065-196 airfoil at different AOAs, carried out by Zha et al. (Zha et al., 2006), it was obvious that the CFJ airfoil's circulation is significantly larger than the airfoils baseline and reversed velocity deficit is present at low AOA in the predicted wake profile.

In another paper, Zha et al. (Zha et al., 2007), modeling CFJ0025-065-196 and CFJ0025-065-000 airfoils, observed that the suction occurring on the airfoil suction surface, such as the CFJ airfoil, is much more beneficial than the airfoil with injection only. However, more vigorous mixing, better circulation, a more filled wake, a higher stall angle of attack, less drag, and less energy are consumed by the CFJ airfoil when it has both injection and suction.

Chng et al. (Chng et al., 2009) conducted a Clark-Y Co-Flow Jet airfoil at different jet momentum coefficients, both experimentally and numerically. According to the results, the co-flow-jet airfoil performs better when both injection and suction are utilized concurrently than when each is employed separately. For the measured range, the increase in the integral of the airfoil static pressure distribution ranges from 40% to 100%, depending on the momentum coefficient, which also affects the lift and stall margin.

Dighe and Cater (Dighe and Cater, 2015) used a co-flow jet S809 wind turbine. At the same time, the injection and suction slots were installed at the locations of 6% c and 80% c from the leading edge, respectively, and observed that the lift, stall margin, and drag reduction were all greatly improved. Also, it completely suppressed a massive separation.

Xu et al. (Xu et al., 2015) investigated an S809 airfoil at different flow velocities and jet momentum coefficient levels. They showed that the lift, stall margin, and drag are all greatly improved with CFJ implementation, and the drag is even decreased to below zero. Also, a considerable amount of power can be gained at a relatively low cost in energy loss for



pumping the co-flow jet.

The CFJ minimizes the energy required to pressurize the jet flow by using the leading-edge lower pressure and the trailing-edge more significant pressure. The overall drag even turns negative in the circumstances of jet momentum coefficients of 0.12 and 0.18. According to suction location research, Xu and Zha (Xu and Zha, 2020) found that the best suction slot is at the geometry infection point at 53% chord because it effectively prevents airfoil separation at high AOA. Furthermore, the final design is accepted with the injection location of a 3% chord due to its improved energy efficiency at high angle of attack. Also, the suction slot size of 1% chord and the injection slot size of 0.75% chord is adopted to minimize the power coefficients. Finally, in optimum conditions, it is observed that $C_{Lmax}$ has increased by 42.3% compared to the baseline S809 airfoil.

Finally, Shi et al. (Shi et al., 2021a) recently simulated a CFJ S809 airfoil HAWT in which the injection port is located at 0%, 0.413%, 1%, 3%, and 5%, and the inhalation port is located at 80%. The injection and inhalation heights of 0.6% and 1.2% of the chord length, respectively, improved HAWT's net torque and power output, which is noticeably higher than the power consumed by the CFJ. The optimum injection port is located at 1% blade chord length. When the CFJ momentum coefficient is 0.029, and the length of the CFJ slot is 4 m (blade spanwise from 20% to 99.4%), the use of the CFJ flow control strategy in a HAWT appears to be the most effective at reducing the range of flow separation and thereby raising the wind turbine's overall efficiency.

### 3.3. plasma actuators

The plasma actuator consists of two thin copper electrodes arranged asymmetrically separated by a dielectric material and wired at a significant voltage difference. One of the electrodes is exposed to the surrounding air, and the other is embedded in a dielectric material (see Fig. 19) (Jayaraman and Shyy, 2003).

Physically, the plasma actuator produces a wall jet on the surface, acting as a source of external momentum to the fluid with a zero-mass flux. When AC voltage is applied, a plasma discharge appears on the surface above the encapsulated electrode, and directed momentum is coupled into the surrounding air (List et al., 2003; Orlov and Corke, 2005).

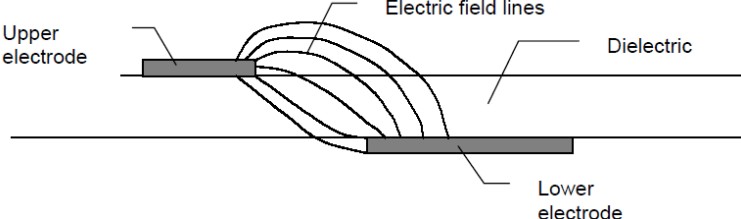

**Fig. 19. The plasma actuator**(Jayaraman and Shyy, 2003)

In the case of the stalled airfoil, the NACA 0015 airfoil is subjected to a flow with a 15 deg angle of attack and a Reynolds number of 45,000 by Gaitonde, D. v, et al.(Gaitonde et al., 2005) . The airfoil is in the stall state, and the plasma actuator is placed close to the leading edge and the trailing edge at different simulations. Compressible Navier-Stokes equations were used with high accuracy, and the plasma force distribution was assumed to be linear (see Fig. 20).

An appropriate amount of this force in the flow direction helps reduce the stall so that the stall angle is increased from 15 to 21 degrees, and complete elimination of the stall in sufficient amounts of plasma force has been observed. However,





the plasma actuator size is smaller than the airfoil chord length.

The 2D and 3D model simulation results are precisely the same when the stall is eliminated. However, with the

presence of the stall, due to the absence of spanwise breakdown vertices, 2D results are different from the 3D model.

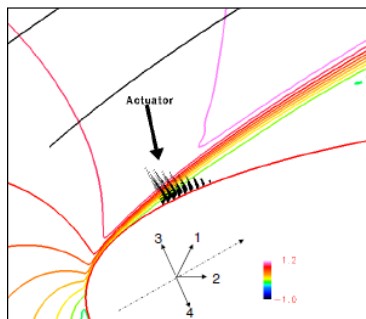

**Fig. 20. Body force field configurations for phenomenological approach** (Gaitonde et al., 2005)

In an investigation by Hikaru Aono (Aono et al., 2013) the S825 wind turbine airfoil is under a flow with a Reynolds number of 760000 and an AOA of 22.1 degrees, which causes severe separation near the leading edge. Also, the lift-to-

drag rate increased from 2.25 to 6.52. However, increasing the momentum of plasma prevents the separation of the boundary layer near the leading edge. However, more than the conditions are required to eliminate the separation completely. In addition, a DU 91-W2-250 airfoil is subjected to flow at different angles of attack, while on the airfoil, one or two plasma actuators are placed in different chord lengths (Mazaheri et al., 2016). The results indicate that the separation area is effectively reduced or eliminated.

Using several plasma actuators also shows that the lift coefficient increases up to 160%, the separation is delayed, and the stall angle increases to 28 degrees. Fig. 21 also displays the effect of plasma types on the maximum lift coefficient and stall angle.



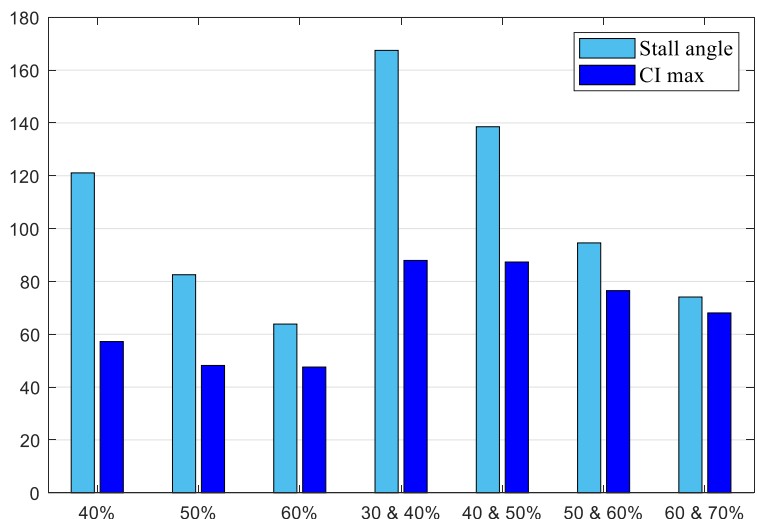

**Fig. 21. effect of single/tandem actuators on the relative increase in the maximum lift coefficient and the stall angle**
**compared with the clean airfoil (in percent)** (Mazaheri et al., 2016)

Also, the number of plasma actuators used on the airfoil is one of the elements of particular importance. A 5M-W
NREL wind turbine with 4,3 and 2 plasma actuators improves the turbine's power (Ebrahimi and Movahhedi, 2017).
The closer these operators are to the center of the turbine, the more the speed increases. Moreover, Increasing the number
of plasmas improves the speed of the turbine. Also, it increases the production of aerodynamic forces, torque, and rotor
output power in all cases. The increase in output power will be 0.85%, 0.77%, and 0.66%, respectively.

NACA 0018 airfoil with different flow speeds of 30 and 65 m/s, Reynolds numbers of 330000 and 715000, and 0 to
20 deg AOA (with the interval of 2), while five and three plasma actuators are placed on the suction side and the pressure
side of the airfoil respectively, have been examined in both numerical and experimental studies(Hoeijmakers and
Meijerink, 2011).

The pressure side actuators increase the lift coefficient of the airfoil at mid-angles (counter flow). Also, the suction
side actuators almost eliminated the separation. However, this effect is reduced for higher angles of attack and speeds.

In general, it can be concluded that plasma actuators can eventually be used as long as the flow has a Reynolds number
of 330,000 and thicker dielectrics with newer materials are required. As the size and material matter, ainJavad Omidi et
al. (Omidi and Mazaheri, 2020) study, a DU21 wind turbine with a plasma near the leading edge (0.04 chord length) is
735 investigated to realize the effects of the size and material of each actuator component on the aerodynamic parameters of
the airfoil. Increasing the length of the embedded electrode and the conductivity of the electrode has limited maximum
performance. Increasing the electrode and dielectric thickness (20-23) reduces the plasma effect.

Jakirlic et al. (Jakirlic, 2019) the positive effects of plasma mounting on the NACA 0012 airfoil. This paper examines
three near-wall turbulence models, the best of which was the transition-sensitive KKLRSM model.

The pulsed plasma jet is another parameter whose change can significantly affect the flow over the airfoil and its



performance. The NREL S822 wind turbine is used in three-dimensional pulse mode in a flow with a Reynolds number of 100000 with an AOA of 5 degrees (Gross et al., 2010). The separation changes are investigated while the plasma actuator is in the thickest part of the airfoil suction side. The results indicate that using a pulsed plasma actuator is effective in wind turbines. Separation is significantly reduced, the lift value is increased by 30%, and the amount of drag is reduced.

Li Guoqiang et al. (Guoqiang and Shihe, 2020) examine a wind turbine with plasma on the leading edge. The difference between pulsed and continuous plasma actuators has been investigated, and it has been concluded that pulsed plasma, in addition to energy-saving, can also have more beneficial effects.

In weak stalls, the actuator easily controls the transition of the boundary layer. It increases the momentum mixing with the flow. At the same time, in higher stall states, the actuator moves the separation location backward and avoids the presence of large vortices in the dynamic stall.

After turning on the plasma, a negative pressure bulge site created by the plasma can return unstable vortices to the surface of the airfoil, which ultimately leads to an increase in lift. (The number of suction peaks increases at the leading edge, and the reverse pressure gradient is recovered near the leading edge, which increases the lift.)

One study [140] examines a vertical axis turbine with different flow speeds while the plasma is installed individually in different parts. Finally, it concluded that pulse plasma is suitable for vertical axis turbines and is maximum at angles of 60-120 degrees.

Some studies are carried out to determine the most efficient direction of a plasma jet on different wind turbine blades. Maria Grazia De Giorgi et al. (De Giorgi et al., 2020) examine the oscillating NACA 23012 turbines, with three micro-plasmas mounted on top and three on the bottom surface. The effect of plasma jet flow directions (co-flow or counter-flow) on the airfoil lift indicates that at the airfoil's upper surface, the plasma jet in the same direction as the flow decreases the pressure and increases the lift. It also increases the lift at the pressure side when the plasma jet is counter-flow. The momentum coefficient is also specified at different attack angles. Also, different on/off plasma methods were also performed to reduce the airfoil loads and increase its stability. The obtained results showed that the turbine is more resistant to fatigue.

In conclusion, micro-plasmas can reduce the unstable loads on the oscillating airfoil, which increases the stability of the airfoil. To optimize the aerodynamic performance of a wind turbine airfoil at Delft University (DU) in a full stall condition, Javad Omidi and Karim Mazaheri1 (Omidi and Mazaheri, 2022) used the operational parameters (voltage, frequency, and waveform) applied to the plasma actuator as the primary design variables. The study found that increasing the applied frequency up to a specific limit enhanced the aerodynamic efficacy of the wind turbine airfoil but that higher frequencies had no discernible effect. In addition, increasing the voltage resulted in a continuous enhancement of the aerodynamic performance, with a 130% increase in the lift coefficient observed. Finally, the study examined three various waveforms for the applied voltage and found that the rectangular waveform generated a more significant lift coefficient.

Also, Philippe Versailles et al. (Versailles et al., 2011) used the DU 96-W-180 two-dimensional wind turbine in an experimental and numerical study. The angle of attack is about zero to 15 degrees, and the flow velocities are 12.6 and 16.30 m/s. At the same time, the plasma actuator is located at the suction side and upstream to investigate the separation of the boundary layer. The result showed that the plasma actuator increases the thickness of the boundary layer, and its



separation, and decreases the lift coefficient.

The frequency used in the problem is 9400 Hz, and the probability of increasing the effects increases with increasing frequency. The aerodynamic performance of a two-blade Savonius VAWT is studied statistically by Xu et al. (Xu et al., 2022) to determine the impact of plasma flow control. Variations in wind turbine efficiency as a function of excitation factors are investigated. According to the findings, placing the plasma excitation at the optimal location in the center of the convex side of the blade significantly boosts the Savonius VAWT's effectiveness. Furthermore, the electric field's force distribution in various directions will modify its effective range. With plasma as its driving force, the wind turbine's overall performance has grown dramatically, with a maximum net power coefficient improvement of about 43.836%.

The pitching and flapping movements of the NACA0012 Airfoil and a flow with a Reynolds number of 12000 are investigated (Mahboubidoust et al., 2017). The plasma actuator is located on the leading-edge and trailing edge. According to the results, it is clear that the plasma on the trailing edge has better effects than the leading edge. Also, this plasma has no positive effect on the flapping movements. Kayvanpour et al. (Kayvanpour et al., 2023) examined the aerodynamic performance of a pitching NACA 0012 airfoil with DBD plasma actuators. The airfoil oscillated beyond the static stall angle in a deep dynamic stall. The plasma actuator increased lift, decreased drag, reduced lift coefficient hysteresis, and delayed separation. The leading edge exhibited the largest stall angle and greatest aerodynamic efficiency, consistent with prior investigations. The research also discovered that increasing voltage and frequency caused a 0.14-degree stall angle delay.

In the paper (Fukumoto et al., 2016), a NACA 633-618 airfoils is used when pitching at -15 to 15 deg angles, and the Reynolds number of the flow is 84000. The plasma actuator is installed at 5%, 10%, and 60% of the chord length. Plasma installation improves aerodynamic performance, and the third model is in the best position in case of increasing $C\_L$ and $L/D$. It improves aerodynamics at 10% but worsens it in some other angles.

Using dielectric barrier discharge (DBD) plasma actuators at low tip speed ratios combined with blade rotations and oscillations performance, Amine and José (Benmoussa and Páscoa, 2023) looked into improving the efficiency of a cycloidal self-pitch vertical axis wind turbine (VAWT). Compared to a conventional fixed-pitch vertical-axis wind turbine, the power coefficient is improved by 38% when the DBD plasma actuation is used with the control law.

Mohaddeseh Fadaei et al. (Fadaei et al., 2021) consider a horizontal axis turbine with a single-plasma actuator at the leading edges, 0.02 and 0.15 chord length. It is evident that the plasma voltage, frequency, angle of attack, and free-flow velocity affect the airfoil's effectiveness. At Reynolds 285000, for example, the maximum impact at 21 degrees of attack angle is 312%, 307%, and 256% at the leading edge, 0.02 and 0.15 chord, respectively. A linear relationship exists between the aerodynamic parameters and the frequency at all voltages.

In addition, the plasma actuator's lift and drag depend on the flow velocity and angles of attack. So that when the plasma actuator is installed on the leading edge, the effectiveness in the Reynolds number of 285,000 at a 21-degree angle of attack was 312%, while in the Reynolds number of 427,000 was reduced to 166%.

A horizontal axis turbine with three NACA0012 airfoils is displayed at a 10 degrees angle of attack with a Reynolds number of 133333 (Aono et al., 2014). The plasma actuator is located on the leading edge. The LES method has also been used to model turbulence. Approximately 11% to 14% increase in torque was observed in the first cycle after plasma installation. This actuator also delays the separation of the turbine's leading edge.

Hikaru Aono et al. (Aono et al., 2020) evaluate a horizontal axis turbine with a plasma actuator on the leading edge



for its average torque. The results showed that the average axial torque increased from 11% to 19%.

K. Asada et al. (Asada et al., 2015) use the NACA 0015 airfoil with a Reynolds number of 63000 and an AOA of 12 degrees. The actuator is located at a 5% airfoil chord length at the suction surface, and the LES method is used for unstable simulation. Results indicate that Lift and drag values have three time zones:

1- Immediate reduction of lift and drag values

2- Lift recovery and increase area

3- Convergence stage of lift and drag values

An investigation has been done by A. Gross et al. (Gross and Fasel, 2012) to study unstable loads on the airfoil. An S822 air wind turbine at an angle of attack of 5° and a Reynolds number of 100000 in three dimensions where the plasma placed on the suction surface reduces unstable loads on the airfoil, which means that the turbine will have a longer life. Also, the drag-to-lift ratio increases from 6.6 to 30, and the separation is effectively reduced. This paper uses three actuators, a vortex generator jet, a flip-flop jet, and a plasma actuator. The most effective and efficient control was achieved for blowing ratio = 0.1.

Various articles in the literature have always emphasized the positive effects of installing a plasma actuator on air turbine airfoils, and many conditions have been studied over the years. Installing this actuator on the suction side in the direction of flow and on the pressure side counter-flow will perform well, increase the lift coefficient, reduce the drag coefficient and cause a delay in separation, and increase the stall angle. Also, increasing the number of plasma actuators, Despite the need for more voltage, has returned better results. In addition, using different pulse methods and being more aerodynamically functional than continuous plasma jets also requires less power consumption. This method also reduces the creation of unstable loads and increases the turbine life by preventing the fatigue phenomenon.

**3.4. synthetic jet**

Synthetic jet actuators (SJA), also known as zero-net-mass-flux actuators (ZNMFA), are innovative jets based on high-frequency blowing and suction, typically via a cavity and a piezoelectric membrane. They require meager power and room, making them very practical for aerodynamic and turbo-machinery devices (Jabbal and Jeyalingam, 2017; Zhong et al., 2007). As mentioned, an SJA comprises a cavity with an orifice on one side and an oscillating membrane on the other (see Fig. 22). It is worth mentioning that the SJAs could be piston type, piezoelectric type, electrodynamic type and plasma type based on the structure of the oscillating membrane (Zhu et al., 2019). The alternative motion of the membrane expels the air through the orifice into the external flow field and entrains the surrounding air back through the same orifice. In the ejection phase, a shear layer is formed and induces a vortex ring due to flow separation near the orifice edge. By the time the membrane moves downward, the vortex ring has moved far enough away not to be affected by the entrainment of the ambient air into the cavity. Thus, even though there is no net mass flow into or out of the cavity during the process, a non-zero mean momentum flux penetrates high-momentum fluid into the boundary layer. For better understanding, the detailed contour of pressure on a controlled and uncontrolled airfoil performed by large-eddy simulation (LES) is depicted in Fig. 23. Therefore, they are suitable for many applications, including wind turbines.

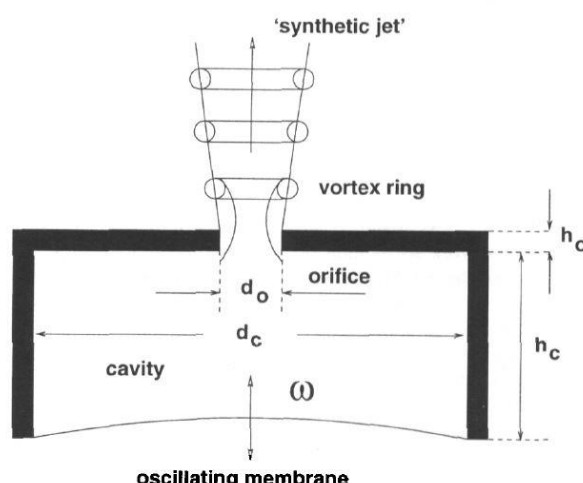

**Fig. 22. Schematic of a synthetic jet actuator structure** (Mallinson et al., 2001)

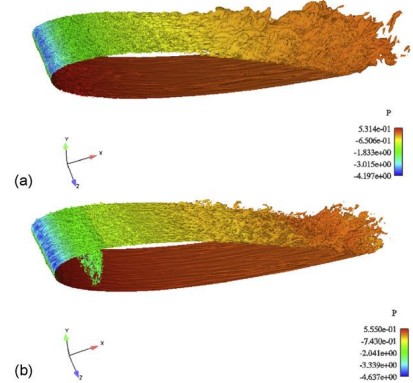

**Fig. 23. Pressure contours: a) uncontrolled case b) controlled case** (You and Moin, 2008)

However, most flow control techniques are challenging in apparatus; synthetic jet actuators were tested experimentally (Amitay et al., 1998, 1999; Rediniotis et al., 2002; Seifert et al., 1993). In the past decades, numerical methods have become one of the most powerful options for investigating different parameters without the cost of constructing experimental structures. Many studies have shown the capability of CFD by validating the experimental results (Lopez et al., 2009; Zhu et al., 2018c).

In 2006, Duvigneau and Visonneau performed a study to optimize the velocity amplitude, frequency, and angle concerning the wall in incidences between 12-24 degrees (Duvigneau and Visonneau, 2006). Naca 0015 airfoil was chosen as the baseline, and a synthetic jet located at 12% of the chord from the leading edge was considered. Moreover, a multi-directional search algorithm developed by Torczon (Torczon, 1989) was used in the optimization procedure. The results show a considerable increase in the maximum lift by 34% and stall delay from 19 to 22 degrees concerning the

initial control parameters. A year later, another study was carried out to find the optimal location of the synthetic jet on the suction side of the NACA 0012 airfoil at 18 and 20 incidences (Duvigneau et al., 2007). The results clearly exhibit





the effect of synthetic jet location on lift changes; at 20 degrees, the lift experiences sudden variation as the jet position changes, while at 18 degrees, it has a smooth trend. According to the results, the lift increment in the optimal state at 18 and 20 degrees is 5% and 57%, respectively, which indicates the sensitivity of the jet's position on performance. The

result was confirmed by Wang & Wu's research on a pitching NACA0012 airfoil (Wang and Wu, 2020). This study showed that although both drag and lift coefficient increase when the jet moves toward the trailing edge, the best aerodynamic performance is achieved by placing the jet near the trailing edge, considering the lift-to-drag ratio. In recent years, Cao et al. (Cao et al., 2020) have described the influence of synthetic jets on fluid behavior in post-stall attack angles from a Lagrangian point of view. In contrast, in most studies, the Eulerian viewpoint has been used. The effect

of jet velocity and frequency on lift enhancement was also investigated. For SJ, the velocity corresponding to the momentum coefficient is defined as Eq. (1.

$$C_\mu = h\sin(\theta)U_{jet}^2/(cU_0^2)$$ (1)

Obviously, the higher amplitude of jet velocity causes a higher momentum coefficient. Thus, four different cases have been analyzed at different velocities(Cao et al., 2020). The momentum and lift coefficient shows a direct behavior since, in the suction phase, the increase in momentum coefficient decreases the pressure coefficient at the airfoil's leading edge

by increasing fluid velocity and the boundary layer momentum in the blowing phase, which delays the separation.

Like the previous study, four cases with different jet frequencies but the same velocity were analyzed. The higher the frequency, the lower the lift coefficient. The graph of the pressure coefficient along the airfoil surface could well explain its reason. All cases show a similar pressure distribution from the leading edge to the middle section. However, the pressure coefficient keeps lower in the separation zone with lower frequency because the energy carried by the jet is

higher in each period.

The importance of horizontal axis wind turbine (HAWT) efficiency provoked Moshafeghi & Hur (Moshfeghi and Hur, 2017) to select an S809 airfoil and carry out a numerical investigation on the effects of SJs on its performance at a wide range of angles of attack from an attached flow to highly separated conditions. The results reveal that SJs negatively affect the lift coefficient at small AOA because the injected flow pushes the attached flow upward and causes an early

separation. On the other hand, when the AOA exceeds the separation onset, SJs could considerably enhance the lift coefficient. Furthermore, the investigation of the effect of jet angle on the performance was conducted in three cases of 5, 15, and 25 degrees. The results did not show any specific pattern, and it caused a 13%, 7%, and 11% improvement, respectively. In 2018, Zhu et al. (Zhu et al., 2018c) studied the impact of the number of orifices on the flow field considering a straight-bladed vertical axis wind turbine (SB-VAWT). The SB-VAWT with 2, 4, and 6 orifices were

tested. It is proved that the power coefficient increases in each case compared to the clean airfoil. Plus, synthetic jets at the leading edge and middle of blades significantly decrease the power coefficient increment. Recently, Wang et al. (Wang et al., 2022) developed an innovative idea and proposed a dual synthetic jets actuator (DSJ) arranged at the trailing edge on a straight-blade vertical axis wind turbine (SB-VAWT) with three arrangement modes. Among all three types, Juxtaposition arrangement mode shows the best improvement of power capacity of VAWT and its highest effect

was achieved when f=360Hz and $C_\mu$=1.975*10^-2 resulted in a 58.87% increase in $C_p$ when TSR=2.05.

In conclusion, the studies show that the performance and efficiency could be affected by jet parameters like speed, frequency, location, etc., and the flow condition. Synthetic jets with a higher momentum coefficient and lower frequency near the leading edge could significantly influence, especially in highly separated conditions.





### 3.5. Active Trailing Edge Flap

Active trailing edge flaps (ATEFs) have been studied extensively to improve wind turbine blades' aerodynamic performance. ATEFs are typically controlled by a feedback system that adjusts the flap angle to optimize the angle of attack and minimize the effects of turbulence and flow separation. Furthermore, as detailed in (Barlas and Van Kuik, 2007), many attempts have been made to build active trailing-edge flaps for wind turbines to decrease vibrations. Because it has to alter the entire blade, traditional pitch control has a response lag and demands a substantial energy

input. Using an ATEF allows for a simple control approach to change the blade's aerodynamic coefficients by actuating the ATEF to various flap angles, reducing fatigue loads, and protecting the blade (Heinz et al., 2011). Many studies on ATEFs have been performed worldwide in the recent decade, and numerous ATEF design concepts have been proposed. The impact of active trailing-edge geometries on aerodynamic performance has been studied in terms of their dimensions and morphologies. In (Sun et al., 2017), a smart material–based active external trailing-edge flap for wind turbine blades

was developed. During normal operation, the potential for vibration fatigue load reduction was assessed. The external flap can lower the variability of the blade root flap wise bending moment and the amount of the damage equivalent loads, according to the fatigue analysis (Oltmann et al., 2017).

Using TEFs with a static deflection affects the mean value and does not affect the bending moment variation. To identify a configuration that eliminates cyclic moment fluctuations, a 1P sinusoidal deflection of the flaps was

introduced. Several simulations were carried out, with maximum flap amplitudes ranging from 0° to 10° and azimuthal phase changes ranging from 0° to 360°. According to this analysis, when the TEFs follow a sinusoidal motion with a phase shift of 49.1° and a flap deflection amplitude of 3.7°, the turbine shows no cyclic 1P fluctuation in the root bending moment. This applies to a TEF with a 20 percent chord depth ranging from 80.5 percent R to 92.7 percent R and a 20 percent chord depth ranging from 80.5 percent R to 92.7 percent R.

A new blade profile design with a fixed trailing-edge flap was examined by Mansi (Mansi and Aydin, 2022b), using a blade element momentum (BEM) approach and confirmed using a computational fluid dynamics (CFD) approach to improve the performance of small-scale HAWTs. The research indicates that the modified blade profile's power coefficient is 8.3 percent greater than the reference design at a design tip speed ratio of 7.

Corsini et al. (Corsini et al., 2015) performed a numerical study on the deformation of the trailing edge flap of the

airfoil in both passive and active modes. They used an FSI code and confirmed the results of previous studies. In the active state, the morphed airfoil has a higher aerodynamic efficiency due to the greater lift coefficient. Still, in the passive mode, the morphing airfoil has a lower drag than a rigid section, which can increase blade torque.

To optimize the flow field around the VAWT, Yang et al. (Yang et al., 2017) have created an airfoil with a trailing edge flap based on the NACA0012 airfoil. The flap angle is adjusted using a flap control strategy, and the results

demonstrate that the flapped airfoil positively impacts damping trailing edge wake separation, postponing dynamic stall, and lowering oscillation amplitude. During the pitch-up interval of the airfoils, the circular wake vortices transform into strip vortices. Flap control improves the dynamic stall by reducing the trend of flow separation according to flow details around the spinning airfoil. Because an oscillating flap constricts the range of the nominal angle of attack, airfoil stall separation is reduced.

Bofeng et al. (Bofeng et al., 2018) also proposed a trailing-edge flap that adopts a Triangle gurney flap with a vortex angle of 60° to reduce the drag force , as illustrated in Fig. 24. The results illustrate that the lift coefficient and the lift-





to-drag ratio of the trailing-edge flap are markedly more extensive than those of the traditional flaps. The trailing-edge flap can effectively improve the stall characteristics of the trailing-edge airfoil. The effects of the flap gap (the gap between the proposed TEF and airfoil body) and deflection angle were also studied. When the separation is bigger than

1 mm, the pressure coefficient drops. Because it has a detrimental impact on stall performance, the distance should be kept as little as possible. In large flap deflection angles, the lift-to-drag ratio increases while the angle of attack is slight; however, when the angle of attack is more than 5°, the lift-to-drag coefficient decreases.

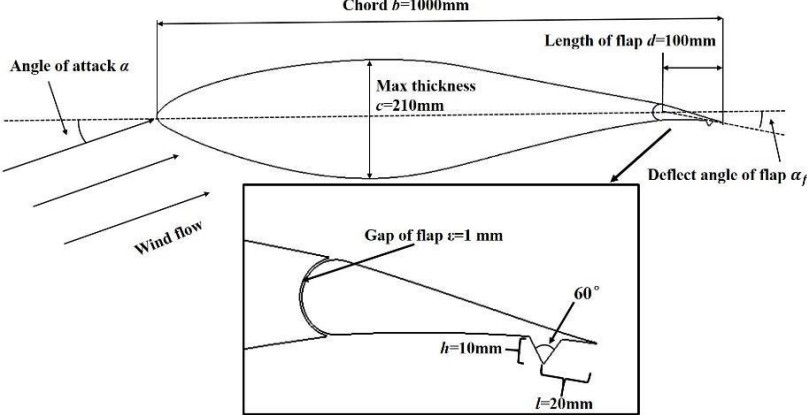

**Fig. 24. Modeling of the S809 airfoil with the proposed trailing-edge flap** (Bofeng et al., 2018)

**4. Discussion**

**4.1. Geometry's Impact on wind turbines**

Overall, studies demonstrate that the geometry of flow controllers can have a significant impact on the performance of wind turbines. By carefully selecting and optimizing the geometry of these devices, wind turbine designers and engineers can achieve significant improvements in efficiency and power output. However, it is important to note that the

optimal geometry may vary depending on the specific wind turbine design and operating conditions.

The effectiveness of passive flow controllers depends heavily on their geometry, which includes several parameters such as length, thickness, width, shape, and location. Increasing flap length and thickness or increasing the slat height, chord length, or cavity width led to a significant increase in lift, while drag increased only slightly. However, increasing the length, thickness, or height beyond a certain point resulted in diminishing returns, indicating that there is an optimal

size for these flow controllers.

Drag will be reduced significantly by increasing the cavity depth, vortex generator's height and spacing, and slot height. Also, it should be noted that there is an optimal size for the cavity, beyond which the performance benefits diminish. Also, the semi-circular shape led to the best performance improvements among different cavities, vortex generators, and slat shapes. This is attributed to the fact that the semi-circular shape had the highest aspect ratio, which

allowed                    for                    greater                    flow                    control.
In addition to these parameters, several others have investigated the effects of the location and orientation of flow controllers on wind turbine performance, which may lead to a significant reduction in drag and an increase in lift. The



effects of vortex generator location and orientation on wind turbine performance. For example, a study investigated the use of a row of vortex generators placed at the leading edge of a wind turbine blade, and some researchers proposed placing a slat at the leading edge.

### 4.2. performance of flow controllers at different TSR

TSR is the ratio of the tangential speed of the blade tip to the wind speed. It is a critical parameter in wind energy systems that affects wind turbines' efficiency, design, safety, and control. A better understanding of TSR can lead to improved performance and reliability of wind turbines, which can help to accelerate the adoption of wind energy as a sustainable energy source.

The effectiveness of **passive flow controllers** can vary depending on the wind turbine's tip speed ratio (TSR). At low TSRs, passive flow controllers can improve the aerodynamic performance of wind turbines by reducing the turbulence intensity and delaying the onset of stall. Stall occurs when the angle of attack of the blade becomes too steep, causing the airflow to separate from the surface of the blade and reducing its lift. By delaying the onset of stall, passive flow controllers can increase the lift generated by the blade and thereby increase the power output of the turbine. At high TSRs, passive flow controllers can also improve the aerodynamic performance of wind turbines by reducing the drag on the blade by manipulating the flow of air around the blade and minimizing the creation of turbulence.

However, the effectiveness of passive flow controllers at different TSRs depends on the design of the controller, the Reynolds number of the flow, and other factors. The Table 2 shows different passive flow controllers have better performance at which range of TSRs.

Table 2.  Performance of passive flow controllers at different TSRs

| flow controller type | work better at | The typical regime of TSR | other factors that effect on optimal TSR | |
|---|---|---|---|---|
| gurney flap | higher TSR | between 7-9 | blade pitch angle and the wind speed. | Lennie et al. (2017) |
| Fixed trailing edge flap | lower TSR | 3-6 | blade twist angle and the wind speed. | Shen et al. (2019) |
| microcylinder | intermediate TSR values | 5-8 | the spacing between the microcylinders and the blade surface, | Arjomandi et al. (2015) |
| cavity | intermediate TSR | 4-7 | the cavity size and spacing | Marzocca et al. (2019) |
| vortex generator | intermediate TSR | 3-6 | the generator size and spacing | Guo et al. (2019) |
| j- type blade | intermediate TSR | 3-8 | the twist angle | Ma et al. (2018) |



| leading edge slat | intermediate TSR | 5-8 | the specific design of the wind turbine and the size of the slat. | Wang et al. (2019) |
| slot | intermediate TSR | 5-9 | the size of the slot | Islam et al. (2019) |

Low TSR regime: In this regime (TSR < 2), the wind flow is relatively smooth and attached to the blades, and AFCs are ineffective. In this case, passive flow control methods improve the turbine's performance more effectively.

In a moderate TSR regime, the wind flow is still relatively attached to the blades, but the flow becomes more complex and unsteady. AFCs that inject air or fluid into the boundary layer, or use suction to remove fluid from the boundary layer, can delay the onset of separation and reduce drag, thereby improving the turbine's performance.

In a high TSR regime, the wind flow becomes highly turbulent and difficult to control. Passive flow control methods, such as blade pitching and yawing, are more effective in improving the turbine's performance in this regime.

In summary, AFCs are most effective in the moderate TSR regime, where they can improve the aerodynamic performance of wind turbines by delaying separation and reducing drag. However, the effectiveness of AFCs depends on the specific design of the controller, the flow conditions, and the particular application in question. Careful analysis and optimization of AFC design are required to achieve the desired performance improvements.

**Table 3. Performance of active flow controllers at different TSRs**

| flow controller type | work better at | The typical regime of TSR | |
|---|---|---|---|
| Microtab and microjet | high TSR | 6 - 9 | Xu et al. (2015) |
| Blowing and Suction | lower TSR | 4-5 | Tescione et al. (2015 |
| Coanda jet | all TSR | 4-10 | Sezer-Uzol et al. (2012) |
| Co-flow jet | high TSR | 6-10 | by Gandhi et al. (2014) |
| plasma actuators | high TSR | 6-7 | Kumar et al. (2017) |
| synthetic jet | high TSR | 6-8 | Lalla et al. (2015) and Ye et al. (2018). |
| Active Trailing Edge Flap | high TSR | 6-8 | Doolan et al. (2015) |

## 5. Conclusions

One of the most effective ways that flow controllers can increase efficiency is by reducing turbulence and smoothing the airflow around the blades. Turbulence can cause drag and reduce lift, which decreases the power output of the turbine. Passive flow controllers such as vortex generators, and winglets can all reduce turbulence and improve lift. Active flow controllers such as boundary layer suction can also reduce turbulence by removing a thin layer of air from the surface of the blade.



Another way that flow controllers can increase efficiency is by optimizing the angle of attack of the blades. Active flow controllers such as blade pitch control and trailing edge flaps can adjust the angle of attack to optimize performance based on changing wind conditions.

    Flow controllers can also increase efficiency by reducing the loads on the blades, which can extend their lifespan and reduce maintenance costs. Active flow controllers such as boundary layer suction can reduce blade loads by removing a

thin layer of air from the surface of the blade.

    In conclusion, flow controllers can increase efficiency at wind turbines by reducing turbulence, optimizing the angle of attack, and reducing blade loads. Both passive and active flow controllers have been shown to be effective in improving turbine performance, and further research is needed to optimize their design for different wind conditions and turbine configurations. The use of flow controllers has the potential to improve the cost-effectiveness and sustainability

of wind energy, making it a promising area for future development.

### Author Contributions

Kiarash Kord was responsible for the data curation and formal analysis of some controllers, ensuring the integrity and accuracy of the data involved in the study. Amir Noori contributed to data curation and performed the formal analysis of active controllers, providing critical insights into the controller mechanisms. Nahid Hasanpour provided resources

and was involved in writing, reviewing, and editing the manuscript, enhancing its overall quality and scholarly presentation. Ali Heydari conducted formal analysis and was the primary author of the original draft, crafting the initial version of the manuscript. Homayoun Askarpour led the conceptualization and investigation processes, guiding the research direction and methodology. Mehdi Kashfi offered supervision, overseeing the project's progression and providing strategic guidance and leadership. Shayan Pakravan was instrumental in editing the text and figures, improving

the manuscript's clarity and visual communication.

All authors have read and agreed to the published version of the manuscript.

### Competing interests

We, the authors of this review paper titled "Performance Enhancements on Wind Turbines Using Flow Controllers: A Review," hereby declare that there are no conflicts of interest to disclose concerning the content discussed in this review

paper.

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



# Contents