# Peer review of "Performance enhancements on wind turbines using flow controllers: A review"

_Wind Energy Science, 2024_

## Referee Comment (RC2)

The authors present a review on flow control devices for airfoils that are utilised to increase the performance and/or lifetime of wind turbines. A large number of active and passive flow control devices are described and a literature review for each of these is presented. A particular focus on this review was the influence of the tip speed ratio and geometry and placement of the flow control devices on the aerodynamic performance of the airfoils.

This is a very comprehensive review, which incorporates a large number of investigated studies and can therefore serve as good database for other researchers to find relevant related works.

1.
However, due to the detailed descriptions, the paper is difficult to read. The lengthy description of references and their conclusions lead to quite cryptic paragraphs where the red line in the text seems to be completely lost. For example:

*"NACA 0018 airfoil with different flow speeds of 30 and 65 m/s, Reynolds numbers of 330000 and 715000, and 0 to 20 deg AOA (with the interval of 2), while five and three plasma actuators are placed on the suction side and the pressure side of the airfoil respectively, have been examined in both numerical and experimental studies(Hoeijmakers and Meijerink, 2011).*
*The pressure side actuators increase the lift coefficient of the airfoil at mid-angles (counter flow). Also, the suction side actuators almost eliminated the separation. However, this effect is reduced for higher angles of attack and speeds. In general, it can be concluded that plasma actuators can eventually be used as long as the flow has a Reynolds number of 330,000 and thicker dielectrics with newer materials are required. As the size and material matter, ainJavad Omidi et al. (Omidi and Mazaheri, 2020) study, a DU21 wind turbine with a plasma near the leading edge (0.04 chord length) is investigated to realize the effects of the size and material of each actuator component on the aerodynamic parameters of the airfoil. Increasing the length of the embedded electrode and the conductivity of the electrode has limited maximum performance. Increasing the electrode and dielectric thickness (20-23) reduces the plasma effect."*

It is therefore highly recommended to generally cut the number of studies that are summarised, group the remaining and improve the red line for the reader in most of the sections (2.X and 3.X).

In most sections, benefits of the particular flow control device for one or the other type of wind turbine are described. This is often confusing because an explanation why the change of a certain airfoil characteristic is exactly beneficial is very short or difficult to understand. It would be helpful to first explain which aerodynamic characteristics of an airfoil are most relevant for the individual wind turbines (HAWT and others) and how changes of these airfoil characteristics influence to turbine performance. This could be maximum lift force, maximum lift-to-drag ratio, stall angle, low drag over a wide range of angle of attack… This knowledge can then be used easily in the description of the particular flow devices to explain in what sense the performance of the particular wind turbine type is increased due to the application of the flow device. In the same way, the disadvantages could be explained.

2.
It feels that VHAWT are in focus of this work, because may references are related to such turbines. As it is very difficult for the reader to distinguish between potential improvements for HAWT and VAWT, the reviewer suggest to concentrate on VHAWT and leave HAWT out. This would improve the readability. If this is not done, a clear indication what kind of wind turbine is subject of the particular paragraph is needed.

3.

The individual sections (2.X and 3.X) appear lengthy in many cases and the reviewer misses a red line. It is therefore proposed to structure each section into the subsections. 1. Description/working principle of the flow control device, 2. Effects/improvements/shortcomings on airfoil level, 3. Effects/improvements/shortcomings on wind turbine level

4.
The reviewer misses a critical discussion of the individual devices. For example in section 2.1, a number of general sentences like
„They significantly improve the lift-drag ratio and have a slightly positive effect on dynamic stall." can be found. From the text, it seems that flaps generally improve the airfoil performance. However, as many airfoils in various applications are not equipped with such devices, it seems obvious that the described improvements have a ‚price'. Meaning that the airfoil characteristic might be improved on the one hand (e.g. an increased lift coefficient), but another quantity (like the maximum lift or the drag or energy consumption) will suffer from the application of such flow device. It seems that a part, where the compromise that the designer has to find is missing in this description. The reviewer believes that such critical discussion is a relevant part of such a review paper. This applies to most of the sections 2.X and 3.X.

5.
The individual sections (2.X and 3.X) often lack a discussion of contradicting findings.

6.
General: Many expressions like 'improvement of 10, 20 or more than 100% lift force and/or performance' are stated without a critical discussion. This gives the impression that such flow controllers have the potential to increase the power performance of wind turbines drastically. However, as such performance jumps will most likely not happen in practise, a critical assessment of such statements is necessary in order to give the reader an advice on how to deal with this information.

7.
The paper should be reworked to become linguistically correct and understandable.

Futher comments can be found below and in the attached PDF.

Abstract:
The abstract contains three hints on the fact that the geometrical parameters of the flow controllers and the influence of the TSR is in focus of this work. For the reviewer's feeling, this is at least one mentioning too much.

Line 89 and following
Would'nt such adaptation to a certain wind speed require an action of any control system? Or how can this adaptation to varying wind speed ranges work passively?

2.5. J-type blade
It seems to the reviewer that this is more a novel blade shape than a flow controller. As the paper is already pretty comprehensive, the reviewer suggests to leave this part out in order to streamline the rest of the paper.

3.1
The sentence
„A micro tab whose height is on the order of the boundary-layer thickness alters the Kutta condition by modifying the camber…"

is not understood by the reviewer. Please explain how the Kutta condition can be altered.

**3.2**
The reviewer feels pretty overwhelmed by the information extracted from this high number of studies. A restructuring and grouping is highly recommended. It should also be checked if all the references add value to the section or may be omitted.

**3.3**
A more precise explanation on how the plasma actuator influences the flow field would be appreciated. The existing explanation on how this "directed momentum is coupled into the surrounding air" and how this leads to a change of the airfoil characteristics seems not quite clear. In addition, the comments related to 3.2 also apply here.

Tables 2 and 3:
The reviewer finds the approach of this table is too general. The influence of the TSR on the flow situation at the blades is strongly dependent on the turbine design and -more important- the turbine type. The reviewer thinks that it is not possible to conclude from the TSR to a certain flow situation at the blades for very different turbine designs and blades. In consequence, it does not seem to be reasonable to give suitable TSR regions for certain flow controllers that are applicable to all discussed turbine types.

[revised manuscript text omitted]

**Contents**

---

## Author Comment (AC1)

Dear Referee,

We appreciate your comments and suggestions and have used them to improve the article.

Your comments for the article "Performance enhancements on wind turbines using flow controllers: A review" along with their responses are below:

In this drafted manuscript, the authors have performed a review study regarding the performance enhancements on wind turbines using flow controllers. After the detailed glance, the main issues for suggestion can be seen as the following comments:

- A chart of nomenclature, symbol, subscripts, abbreviations should be prepared in terms of better understanding for readers.

Response and action: According to your recommendation, the table has been added to the end of the article.

- The resolutions of few figures are not readable and too small. They must be reedited.

Response: Figures 6, 9, 10, 16, 23, and 24 are of average quality. They have been collected from different references and have been placed in this article with the same quality as in the original reference. The rest of the figures are of good quality.

- Provide comprehensive "Highlights" for the text, indicating the contributions of the paper, as allowed by the Journal.

- The writing of the drafted paper should be revised from top to bottom with an experienced helper.

Response and action: Parts of the paper are revised and changed. The other reviewer has sent us many comments with details, based on which we have changed the article a lot. In sections 2 and 3, paragraphs and sentences are rewritten and moved. Many sections are classified into several subsections.

In addition to comments mentioned above, it is not obligatory but the authors might be suggested to add the current and actual studies in the introduction part including explanation passive flow control methods over different airfoils as the following related and current studies so that the scope of this review study is wider:

- Traditional and new types of passive flow control techniques to pave the way for high maneuverability and low structural weight for UAVs and MAVs
- Investigation of pre-stall flow control on wind turbine blade airfoil using roughness element.

- Investigation of the effect of hidden vortex generator-flap integrated mechanism revealed in low velocities on wind turbine blade flow

- Passive Flow Control Application Using Single and Double Vortex Generator on S809 Wind Turbine Airfoil

- Effect of partial flexibility over both upper and lower surfaces to flow over wind turbine airfoil

The comments and suggestions mentioned above should be taken into consideration carefully for better review paper.

Response:
Our article is a review on the use of flow controllers in wind turbines, which is a wide topic. We have tried to consider important topics and focus on them. This focus led to deep analysis and maintaining a coherent structure throughout the paper. Adding suggested studies expands the topics of the article and reduces the depth of the concepts that were the main purpose of the article. We think the current topics are sufficient for the purpose of our review article.

Thank you for your feedback and comments, which are crucial for enhancing the quality of this article. Your time and attention are greatly appreciated.

---

## Author Comment (AC3)

Dear Referee,
Thanks for your comments and suggestions, we have used them to improve the quality of the article.
Below are your comments with their responses (highlighted in yellow).

The authors present a review on flow control devices for airfoils that are utilised to increase the performance and/or lifetime of wind turbines. A large number of active and passive flow control devices are described and a literature review for each of these is presented. A particular focus on this review was the influence of the tip speed ratio and geometry and placement of the flow control devices on the aerodynamic performance of the airfoils.

This is a very comprehensive review, which incorporates a large number of investigated studies and can therefore serve as good database for other researchers to find relevant related works.

1. However, due to the detailed descriptions, the paper is difficult to read. The lengthy description of references and their conclusions lead to quite cryptic paragraphs where the red line in the text seems to be completely lost. For example:

"NACA 0018 airfoil with different flow speeds of 30 and 65 m/s, Reynolds numbers of 330000 and 715000, and 0 to 20 deg AOA (with the interval of 2), while five and three plasma actuators are placed on the suction side and the pressure side of the airfoil respectively, have been examined in both numerical and experimental studies(Hoeijmakers and Meijerink, 2011). The pressure side actuators increase the lift coefficient of the airfoil at mid-angles (counter flow). Also, the suction side actuators almost eliminated the separation. However, this effect is reduced for higher angles of attack and speeds. In general, it can be concluded that plasma actuators can eventually be used as long as the flow has a Reynolds number of 330,000 and thicker dielectrics with newer materials are required. As the size and material matter, ainJavad Omidi et al. (Omidi and Mazaheri, 2020) study, a DU21 wind turbine with a plasma near the leading edge (0.04 chord length) is investigated to realize the effects of the size and material of each actuator component on the aerodynamic parameters of the airfoil. Increasing the length of the embedded electrode and the conductivity of the electrode has limited maximum performance. Increasing the electrode and dielectric thickness (20-23) reduces the plasma effect."

It is therefore highly recommended to generally cut the number of studies that are summarised, group the remaining and improve the red line for the reader in most of the sections (2.X and 3.X).

Response: In sections 2 and 3, the paragraphs have been rearranged and made smaller. Some references have been removed.

In most sections, benefits of the particular flow control device for one or the other type of wind turbine are described. This is often confusing because an explanation why the change of a certain airfoil characteristic is exactly beneficial is very short or difficult to understand. It would be helpful to first explain which aerodynamic characteristics of an airfoil are most relevant for the individual wind turbines (HAWT and others) and how changes of these airfoil characteristics influence to turbine performance. This could be maximum lift force, maximum lift-to-drag ratio, stall angle, low drag over a wide range of angle of attack… This knowledge can then be used easily in the description of the particular flow devices to explain in what sense the performance of the particular wind turbine type is increased due to the application of the flow device. In the same way, the disadvantages could be explained.

2. It feels that VHAWT are in focus of this work, because may references are related to such turbines. As it is very difficult for the reader to distinguish between potential improvements for HAWT and VAWT, the reviewer suggest to concentrate on VHAWT and leave HAWT out. This would improve the readability. If this is not done, a clear indication what kind of wind turbine is subject of the particular paragraph is needed.

Response: Both HAWT and VAWT turbine types are important for the purposes of this review article, so we decided to keep the material for each and improve the text in Sections 2 and 3.

3. The individual sections (2.X and 3.X) appear lengthy in many cases and the reviewer misses a red line. It is therefore proposed to structure each section into the subsections. 1. Description/working principle of the flow control device, 2. Effects/improvements/shortcomings on airfoil level, 3. Effects/improvements/shortcomings on wind turbine level

Response: According to your comment, we classified and arranged the contents of sections 2 and 3 into 3 subsections for each flow controller:
- Description/working principle of the flow control device
- Effects/improvements/shortcomings on airfoil level
- Effects/improvements/shortcomings on wind turbine level

4. The reviewer misses a critical discussion of the individual devices. For example in section 2.1, a number of general sentences like „They significantly improve the lift-drag ratio and have a slightly positive effect on dynamic stall." can be found. From the text, it seems that flaps generally improve the airfoil performance. However, as many airfoils in various applications are not equipped with such devices, it seems obvious that the described improvements have a ‚price'. Meaning that the airfoil characteristic might be improved on the one hand (e.g. an increased lift coefficient), but another quantity (like the maximum lift or the drag or energy consumption) will suffer from the application of such flow device. It seems that a part, where the compromise that the designer has to find is missing in this description. The reviewer believes that such critical discussion is a relevant part of such a review paper. This applies to most of the sections 2.X and 3.X.

Response: In the classification and arranging of sections 2 and 3, according to the previous comments, the critical discussion related to each flow controller was tried to be clearer and more qualitative. It should be noted that many of the references used in this review article do not give us comprehensive and complete information. It has been tried to write the most coherent text with the available information and provide the necessary details for each flow controller.

5. The individual sections (2.X and 3.X) often lack a discussion of contradicting findings.

Response: In sections 2 and 3, as much as possible from the references, contradicting findings have been written.

6. General: Many expressions like 'improvement of 10, 20 or more than 100% lift force and/or performance' are stated without a critical discussion. This gives the impression that such flow controllers have the potential to increase the power performance of wind turbines drastically.

However, as such performance jumps will most likely not happen in practise, a critical assessment of such statements is necessary in order to give the reader an advice on how to deal with this information.

[Figure]

7. The paper should be reworked to become linguistically correct and understandable. Futher comments can be found below and in the attached PDF.

Response: The text has been modified to some extent and all your comments have been included in the text. Parts of the text that have changed due to the comments in the text are highlighted in yellow.

Abstract: The abstract contains three hints on the fact that the geometrical parameters of the flow controllers and the influence of the TSR is in focus of this work. For the reviewer's feeling, this is at least one mentioning too much.

Response: Based on the comment, the abstract is revised.

Line 89 and following Would'nt such adaptation to a certain wind speed require an action of any control system? Or how can this adaptation to varying wind speed ranges work passively?

Response: In the new version of the article, this paragraph has been moved and changed according to the comment.

2.5. J-type blade It seems to the reviewer that this is more a novel blade shape than a flow controller. As the paper is already pretty comprehensive, the reviewer suggests to leave this part out in order to streamline the rest of the paper.

J-type blade design represents a hybrid approach rather than a conventional flow controller. Its distinctive shape greatly affects the air flow in terms of pressure distribution and stall characteristics. By integrating flow control principles directly into the blade design, the J-type configuration demonstrates an innovative method for enhancing wind turbine performance.

3.1 The sentence "A micro tab whose height is on the order of the boundary-layer thickness alters the Kutta condition by modifying the camber…" is not understood by the reviewer. Please explain how the Kutta condition can be altered.

Response: Using the comment, the paragraph is revised.

3.2 The reviewer feels pretty overwhelmed by the information extracted from this high number of studies. A restructuring and grouping is highly recommended. It should also be checked if all the references add value to the section or may be omitted.

Response: According to previous comments, the article has been restructured and some references have been removed.

3.3 A more precise explanation on how the plasma actuator influences the flow field would be appreciated. The existing explanation on how this "directed momentum is coupled into the surrounding air" and how this leads to a change of the airfoil characteristics seems not quite clear. In addition, the comments related to 3.2 also apply here.

Response: According to this comment, previous comments, and comments in the text, sections 3.2 and 3.3 have been modified.

Tables 2 and 3: The reviewer finds the approach of this table is too general. The influence of the TSR on the flow situation at the blades is strongly dependent on the turbine design and -more important- the turbine type. The reviewer thinks that it is not possible to conclude from the TSR to a certain flow situation at the blades for very different turbine designs and blades. In consequence, it does not seem to be reasonable to give suitable TSR regions for certain flow controllers that are applicable to all discussed turbine types.
?

Thank you for your feedback and comments, which is crucial for enhancing the quality of this article. Your time and attention are greatly appreciated.